# Crosstalk between SUMOylation and ubiquitylation controls DNA end resection by maintaining MRE11 homeostasis on chromatin

Tao Zhang [1], Han Yang[1], Zenan Zhou[1], Yongtai Bai[1], Jiadong Wang[1] & Weibin Wang [1] ✉

DNA end resection is delicately regulated through various types of post-translational modifications to initiate homologous recombination, but the involvement of SUMOylation in this process remains incompletely understood. Here, we show that MRE11 requires SUMOylation to shield it from ubiquitin-mediated degradation when resecting damaged chromatin. Upon DSB induction, PIAS1 promotes MRE11 SUMOylation on chromatin to initiate DNA end resection. Then, MRE11 is deSUMOylated by SENP3 mainly after it has moved away from DSB sites. SENP3 deficiency results in MRE11 degradation failure and accumulation on chromatin, causing genome instability. We further show that cancer-related MRE11 mutants with impaired SUMOylation exhibit compromised DNA repair ability. Thus, we demonstrate that MRE11 SUMOylation in coordination with ubiquitylation is dynamically controlled by PIAS1 and SENP3 to facilitate DNA end resection and maintain genome stability.

DNA double-strand breaks (DSBs) are highly deleterious lesions that when unrepaired can result in gene mutations or cell death[1]. Cells typically use two pathways to repair DSBs: non-homologous end joining (NHEJ) and homologous recombination (HR)[2]. The NHEJ pathway joins the DSB ends directly, whereas HR uses the homologous DNA as a template to perform the error-free repair[3,4]. A primary determinant of repair pathway choice is DNA end resection, which generates 3′ long-range overhang DNA that is the prerequisite for HR. Then, replication protein A (RPA) on single-strand DNA (ssDNA) is subsequently replaced with RAD51 to form nucleoprotein filaments capable of searching and invading homologous DNA for accurate repair[5].

DNA end resection needs to be precisely regulated because insufficient or excessive resection threatens genome stability[6,7]. At the molecular level, the key factors to initiate DNA end resection include the MRE11-RAD50-NBS1 (MRN) complex, CtIP, and the BRCA1-BARD1 complex[8–10]. When a DSB occurs, the MRN complex is rapidly recruited on broken DNA ends. Subsequently, the MRE11 endonuclease-catalyzed resection generates a nick in the 5′ strand, and then MRE11, acting as an exonuclease, resects toward the DSB end in the 3′ to 5′ direction[11]. CtIP, a critical MRN complex co-factor, triggers MRE11 endonucleolytic activity[12]. Meanwhile, the BRCA1-BARD1 complex promotes DNA end resection by antagonizing the 53BP1-RIF1 complex in a cell cycle-dependent manner[13]. To date, a series of post-translational modifications (PTMs), including phosphorylation and ubiquitylation, have been reported to regulate DNA end resection[14–16]. For example, phosphorylation of CtIP is required for the MRN endo-nuclease activity that initiates DNA end resection[17]. NBS1 ubiquitylation by RNF8 promotes NBS1 recruitment to DSBs[18].

In addition, SUMOylation has also been suggested to be involved in DNA end resection[19,20]. In pace with the recruitment of DNA damage response (DDR) factors after DSBs occur, a cascade of proteins participating in SUMOylation, such as SUMO1/2/3, the E2 conjugating enzyme UBC9 and certain SUMO E3 ligases, are rapidly recruited onto the damaged sites[21,22]. Multiple lines of evidence have demonstrated

[1]Department of Radiation Medicine, School of Basic Medical Sciences, Peking University Health Science Center, Beijing 100191, China. ✉e-mail: weibinwang@bjmu.edu.cn

that several key DNA end resection factors are SUMOylated and then degraded through a SUMO-targeted ubiquitin E3 ligase (STUbL) mechanism, which facilitates protein turnover at damaged chromatin[19,23]. For example, both BRCA1 SUMOylation and protein levels were increased upon STUbL RNF4 expression knockdown[24,25]. The downregulation of RNF4 expression also caused BARD1 accumulation at damaged DNA sites[23], indicating that the BRCA1-BARD1 complex is regulated through the STUbL mechanism. In addition, phosphorylated CtIP can be modified by SUMO2, and then SUMOylated CtIP is recognized by RNF4 for its degradation, facilitating its turnover at DSB sites[19]. Previous studies have mainly focused on how SUMOylation promotes protein turnover during late-stage DNA end resection. However, whether SUMOylation is directly involved in the initiation of DNA end resection remains unclear.

In this study, we provide evidence showing that SUMOylation shields MRE11 from ubiquitin-mediated degradation but does not trigger STUbL mechanism activation in the initiation of DNA end resection. Specifically, we find that MRE11 is SUMOylated by a protein inhibitor of activated STAT1 (PIAS1) on chromatin to prevent MRE11 ubiquitylation during DNA end resection. Subsequently, MRE11 is deSUMOylated by SUMO-specific peptidase 3 (SENP3), mainly after it is released into the nucleoplasm from DSB ends. SENP3 deficiency results in MRE11 accumulation and genome instability. Furthermore, we find that cancer-related MRE11 mutants exhibit both ectopic SUMOylation and compromised DNA end resection ability. Our study thus suggests that dynamic SUMOylation of MRE11 as regulated by the PIAS1-SENP3 axis acts in conjunction with ubiquitylation to ensure appropriate DNA end resection and genome integrity.

## Results

### MRE11 SUMOylation is enhanced in response to DNA damage

To explore the function of SUMOylation in the initiation of DSB end resection, we used the SUMO inhibitor ML792, which blocks SUMO-activating E1 enzyme activity, to detect the effect of SUMOylation blockade on the resection marker pRPA2 (S4/S8) and activation of the ATR-CHK1 pathway. First, a sharp reduction in SUMO1/2/3-mediated whole cell SUMOylation was observed after ML792 treatment, and exposure to ML792 alone did not cause DNA damage (Supplementary Fig. 1a, b). We then found that after camptothecin (CPT, a topoisomerase I inhibitor) treatment, pRPA2 protein level and activation of the ATR-CHK1 pathway were attenuated by ML792 in a dose- and time-dependent manner (Fig. 1a, b). Then, the number of pRPA2 immunofluorescent foci in the group treated with a combination of ML792 and CPT was significantly lower than that in the CPT only group in the S/G2 phase (Fig. 1c, d). This finding suggests that suppression of SUMOylation is detrimental to DNA end resection.

Several previous studies suggested that BRCA1, BARD1, and CtIP are affected by STUbL RNF4[19,23]. In this study, we also found that the RAD50 protein level was increased after RNF4 was knocked down in HeLa cells expressing His-SUMO2 (Supplementary Fig. 1c), in accordance with the previous study[23], indicating that RAD50 may also be regulated by STUbL. However, the STUbL mechanism cannot explain our above finding that SUMOylation facilitates the initiation of DNA end resection. Thus, we wondered whether MRE11 and NBS1 are SUMOylated to initiate DNA end resection. Covalent SUMO modification of ectopic SFB (a triple recombinant tag containing an S-protein tag, a 2×Flag tag, and a streptavidin-binding peptide tag)-MRE11 and SFB-NBS1 was detected in cells cotransfected with His-SUMO1/2/3 in denaturing buffer (Fig. 1e, f). Interestingly, the SUMOylation levels of MRE11 and NBS1 followed different trends. MRE11 SUMOylation was dramatically increased in both 293 T and HeLa cells treated with DSB-inducing agents (CPT, etoposide, and cisplatin) but not boosted in those treated with hydroxyurea (HU), which leads to replication fork stalling (Fig. 1g, h and Supplementary Fig. 1d–f). In contrast, SUMOylation of NBS1 was inhibited under DNA damage conditions (Fig. 1i),

and the total SUMOylation level of NBS1 was decreased by CPT treatment in a time-dependent manner (Fig. 1j), supporting that SUMOylation inhibitor suppresses resection initiation unlikely through affecting NBS1 SUMOylation.

Therefore, we focused on MRE11 SUMOylation in further investigations. To this end, purified MRE11 was used in an in vitro SUMOylation assay and was found to be SUMOylated by SUMO1 and SUMO2, and shifted SUMO bands were not detected without ATP and SUMO1/2 being present (Fig. 1k). To investigate SUMO modification of endogenous MRE11, we used an anti-MRE11 antibody in denaturing immunoprecipitation (IP) assay following CPT treatment, and the increase in SUMOylation was consistent with the results of the ectopic expression experiments (Fig. 1l and Supplementary Fig. 1g). Importantly, we found that the majority of SUMOylated MRE11 was detected in chromatin fractions (Fig. 1m), implying that MRE11 SUMOylation may be enhanced after its recruitment to chromatin. Collectively, the above results suggest that SUMOylated MRE11 on chromatin most likely facilitates the initiation of DNA end resection.

### PIAS1 is the primary SUMO E3 ligase to stimulate MRE11 SUMOylation

SUMO E3 ligase can efficiently and rapidly catalyze substrate SUMOylation. Next, we tried to identify the SUMO E3 ligase capable of stimulating MRE11 SUMOylation. Cells were co-transfected with SFB-MRE11, His-SUMO2, and Myc-tagged SUMO E3 ligases, including PIAS1, PIAS2α, PIAS2β, PIAS3, PIAS4, and CBX4, and the positive control SUMO E2 UBC9. Among these ligases, only PIAS1 dramatically promoted MRE11 SUMOylation under normal conditions and after CPT treatment, in a dose-dependent manner (Fig. 2a, b and Supplementary Fig. 2a). In contrast, other SUMO E3 ligases did not lead to the same increase in MRE11 SUMOylation. PIAS4 and CBX4, which have been reported to be the E3 ligases for CtIP SUMOylation[26,27], exerted only a minor effect on MRE11 SUMOylation (Fig. 2a, b and Supplementary Fig. 2b). As shown in Fig. 2c, the catalytically inactive mutant PIAS1-CI lost the ability to stimulate MRE11 SUMOylation. Furthermore, the major enhancement of MRE11 SUMOylation by PIAS1 was detected in chromatin fractions, indicating PIAS1-mediated MRE11 SUMOylation mainly occurs on chromatin (Supplementary Fig. 2c). In cells with PIAS1 knockdown by siRNA, MRE11 SUMOylation was significantly inhibited under both normal and DNA damage conditions (Fig. 2d). An in vitro SUMOylation assay also revealed that SUMO2 modification of MRE11 was greatly enhanced by PIAS1 (Fig. 2e). Furthermore, SFB-MRE11 interacted with endogenous PIAS1, and their interaction ability was not affected by CPT, suggesting that MRE11-PIAS1 interaction is independent of DNA damage (Fig. 2f). Recombinant MBP-PIAS1 and Flag-MRE11 proteins were used in an in vitro pull-down assay, and Flag-MRE11 was found to directly bind to MBP-PIAS1 and vice versa (Fig. 2g and Supplementary Fig. 2d). These results suggest that PIAS1 is a critical SUMO E3 ligase for MRE11 SUMOylation.

PIAS1 is a member of the PIAS family SUMO E3 ligase, and its recruitment to laser-induced DNA damage sites depends on the SAP (SAF-A/B, Acinus, and PIAS) domain, which is regarded as a DNA-binding motif[28,29]. Thus, we performed an electrophoretic mobility shift assay (EMSA) to test the DNA binding ability of PIAS1. We confirmed that PIAS1 protein showed a binding affinity for overhang DNA substrates (Fig. 2h). Considering that MRE11 is also a DNA-binding protein, we thus hypothesized that PIAS1 can further stimulate MRE11 SUMOylation in the presence of DNA. Our results showed that an overhang DNA not only promoted the interaction between PIAS1 and MRE11 (Fig. 2i) but also facilitated PIAS1-mediated MRE11 SUMOylation (Fig. 2j), supporting a model in which broken DNA ends serve as scaffolds for PIAS1 and MRE11 recruitment. In summary, PIAS1 is the major SUMO E3 ligase for MRE11, and DNA contributes to MRE11 SUMOylation by enhancing the interaction between PIAS1 and MRE11 on chromatin.

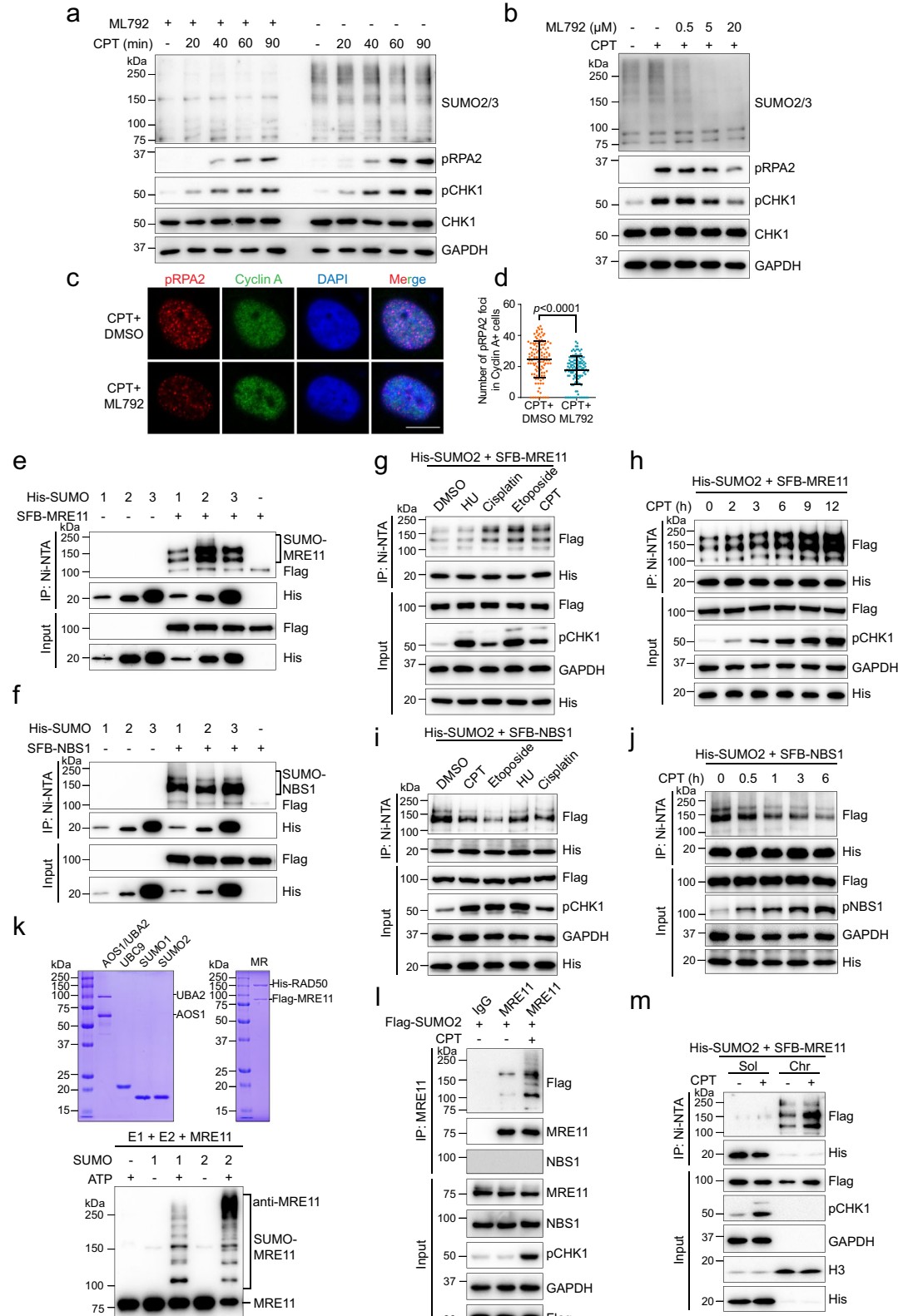

## Identification of MRE11 SUMOylation sites

Next, we sought to map the SUMOylation sites in MRE11. SUMOylation sites are usually identified in two ways. In one method, software is used to predict SUMOylated sites on the basis that SUMOylation is often found in the consensus motif ΨKxE/D (where Ψ represents a large hydrophobic amino acid, and x is any amino acid) or in phosphorylation-dependent SUMOylation motifs (PDSMs)[30,31].

Another method is using mass spectrometry (MS), which provides accurate results but has the technical difficulty of raw data analyses[32,33]. Here, we first analyzed eight MRE11 SUMOylation sites obtained through MS from a previous study:[34] K255, K384, K407, K416, K442, K467, K510, and K625 (Fig. 3a). We replaced every lysine residue with an arginine residue, and found that SUMOylation at four single-mutant lysine sites, K255, K384, K416, and K467, was reduced compared with

**Fig. 1 | MRE11 SUMOylation is increased in response to DNA damage. a**, **b** HeLa cells were treated with or without 20 μM ML792 for 1 h and the indicated times of 1 μM CPT (**a**), or the indicated doses of ML792 and 1 μM CPT for 1 h (**b**) and the whole cell extracts were subjected to immunoblotting. **c**, **d** ML792 repressed pRPA2 (S4/S8) foci formation in S/G2 phase HeLa cells after 1 μM CPT treatment for 45 min. Scale bar, 10 μm. The data are presented as means ± SD, n (DMSO) = 117 cells, n (ML792) = 119 cells. **e** MRE11 was modified by SUMO1/2/3. HEK293T cells were cotransfected with SFB-MRE11 and 10×His-SUMO1/2/3, followed by Ni-NTA pull-down with guanidine denaturing buffer. **f** NBS1 SUMOylation was tested as in **e**. **g** MRE11 SUMOylation was enhanced by treatment with DSB-inducing agents (1 μM CPT for 2 h, 100 μM etoposide for 3 h, and 1 μM cisplatin for 3 h) but not HU (2 mM, 3 h). **h** MRE11 SUMOylation increased after 1 μM CPT treatment in a time-dependent manner. **i** NBS1 SUMOylation was downregulated upon treatment with DNA-damaging agents as in **g**. **j** NBS1 SUMOylation gradually decreased along CPT treatment. **k** The indicated purified proteins were analyzed by Coomassie blue staining, and then used in reconstituted reactions to test MRE11 SUMOylation in vitro. **l** SUMOylation of endogenous MRE11 was examined in HEK293T cells treated with or without CPT (1 μM, 6 h). **m** HEK293T cells expressing SFB-MRE11 and His-SUMO2 were fractionated into soluble and chromatin fractions, followed by the analysis of MRE11 SUMOylation.

that of wild-type (WT) MRE11 (Fig. 3b). Then, we constructed a plasmid carrying MRE11 with all four lysine-to-arginine mutation sites (named 4KR). Unlike 4KR, the mutation of software-predicted lysine sites with high scores (K66, K204, K250, and K496) did not cause an obvious SUMOylation decrease (Supplementary Fig. 3a). It has been reported that MRE11 undergoes ATM-mediated phosphorylation after DNA damage[35]. Thus, we tested whether MRE11 has ATM- and ATR-induced PDSMs. As shown in Fig. 3c, neither an ATM inhibitor (ATMi) nor an ATR inhibitor (ATRi) caused a reduction in MRE11 SUMOylation, indicating that MRE11 SUMOylation is independent of ATM/ATR-mediated phosphorylation after DNA damage.

We found that SUMOylation of the 4KR mutant was dramatically lower than that of WT MRE11, and although some residual SUMOylation was detected in 4KR, it showed only a slight increase after DNA damage or PIAS1 overexpression (Fig. 3d). In addition, 4KR mutation did not affect the interaction between MRE11 and PIAS1 (Supplementary Fig. 3b). To confirm this result, we further purified the MRE11 4KR mutant and assayed SUMOylation levels in vitro. As shown in Fig. 3e, 4KR SUMOylation was greatly decreased compared with that of WT MRE11. Taken together, these data indicate that the four identified MRE11 lysine sites, K255, K384, K416, and K467, are major SUMOylation sites.

### SUMOylation stabilizes MRE11 on chromatin during end resection by antagonizing ubiquitylation

It has been reported that MRE11 is ubiquitylated after DNA damage[36]. SUMOylation may be an underlying mechanism that protects MRE11 from ubiquitin-mediated degradation during DNA end resection. We used the 4KR as SUMOylation-deficient MRE11 to detect the function of SUMOylation when DNA is damaged. As shown in Fig. 4a, 4KR displayed a shorter protein half-life than WT MRE11, and this decrease was more obvious after CPT-induced DNA damage. Each single SUMO site mutant showed an increased degradation rate with different degrees after CPT treatment (Fig. 4a). Meanwhile, in HEK293T cells over-expressing ubiquitin, the 4KR mutant exhibited higher levels of ubiquitylation than WT MRE11 in normal and DNA damage conditions (Fig. 4b). Intriguingly, the WT protein showed more stability after DNA damage treatment, indicating that WT MRE11 resists ubiquitylation under continuous DNA damage stress (Fig. 4b). The K48-linked ubiquitin chain, which mediates proteasomal degradation, dominated the ubiquitylation of 4KR (Supplementary Fig. 4a). Furthermore, PIAS1 knockdown obviously shortened the protein half-life of WT MRE11, and slightly caused a further decrease in 4KR half-life (Supplementary Fig. 4b). We further found that the decreased protein level of 4KR following long-term stress was rescued by the proteasome inhibitor MG132 (Supplementary Fig. 4c).

Next, we explored more details of the interplay between MRE11 ubiquitylation and SUMOylation. HEK293T cells were co-transfected with SFB-MRE11 and His-hemagglutinin (HA) tag-ubiquitin (Ub)/His-SUMO2 to detect Ub- and SUMO-covalent modification at various time points. The MRE11 ubiquitylation level in whole cell and chromatin fractions decreased along with the continuous DSB-inducing drug treatment, and once the stress was relieved, the ubiquitylation level was significantly increased (Fig. 4c and Supplementary Fig. 4d, e). This

fluctuation in MRE11 ubiquitylation was oppositely correlated with that of SUMOylation (Fig. 4d). Moreover, WT MRE11 on chromatin but not in the soluble fractions was degraded rapidly in the presence of a SUMO inhibitor (Fig. 4e). These results indicate that MRE11 SUMOylation antagonizes ubiquitylation to protect MRE11 stability on damaged chromatin during DNA end resection, while MRE11 deSU-MOylation leads to ubiquitylation-mediated degradation.

We further compared the recruitment of WT MRE11 and 4KR to chromatin after DNA damage. The 4KR protein level was lower than that of WT on chromatin under normal conditions. Upon CPT-induced damage, the amount of 4KR recruited to chromatin increased but remained consistently below the WT level at the same time points, and these differences were abolished by MG132 treatment (Fig. 4f). Immunofluorescence showed that the number of MRE11 foci in 4KR was lower than that in the WT after CPT treatment, but there was no significant difference between WT and 4KR in the presence of MG132 (Fig. 4g, h). Therefore, the lower protein level of 4KR on chromatin was caused by its degradation. These findings indicate that SUMOylation-deficient MRE11 cannot maintain a steady presence on chromatin because it is easily degraded.

### MRE11 SUMOylation is important for efficient HR and cell survival

MRE11 plays a pivotal role in DNA end resection via its nuclease activity[37]. Our results showed that cells expressing the 4KR mutant exhibited sharp reductions in the number of pRPA2 and RAD51 foci after CPT-induced DNA damage (Fig. 5a, b). Meanwhile, in shMRE11-cells, 4KR/single-site mutant complement failed to efficiently activate pRPA2 and the ATR-CHK1 pathway (Fig. 5c), suggesting that DNA end resection was inhibited in 4KR/single-site mutant cells. Then, we performed DR-GFP reporter assay to detect HR efficiency. As shown in Fig. 5d and Supplementary Fig. 5a, 4KR exhibited sharply reduced HR efficiency compared with WT cells. Moreover, 4KR mutation rendered cells more sensitive to CPT and cisplatin than WT MRE11 (Fig. 5e-g). These results show that SUMO-MRE11 deficient cells have DNA end resection and HR defects, and thus are hypersensitive to DSB-inducing agents.

MRE11, RAD50 and NBS1 assemble into a complex that is crucial for DNA end resection[38]. Therefore, we performed co-IP assays in cells and in vitro pull-down assays to assess the effect of SUMOylation on the MRN complex assembly. As shown in Fig. 5h, i and Supplementary Fig. 5b, SUMOylation site mutation in 4KR did not impair its interaction with RAD50 and NBS1 either in cells or in vitro, suggesting that SUMOylation of MRE11 is not required for the physical interactions among MRE11, RAD50 and NBS1. Considering that SUMOylation site mutation may change MRE11 nuclease activity, we used purified WT MRE11 and 4KR mutant proteins in reconstituted reactions to evaluate nuclease activity. As shown in Fig. 5j, k, the nuclease activity of 4KR was only minorly reduced compared to that of WT MRE11.

### The stability of SUMOylation-deficient MRE11 is restored upon fusion with a poly-SUMO2 chain

To verify that the protein instability and HR defects associated with 4KR were caused by SUMOylation defects, we constructed

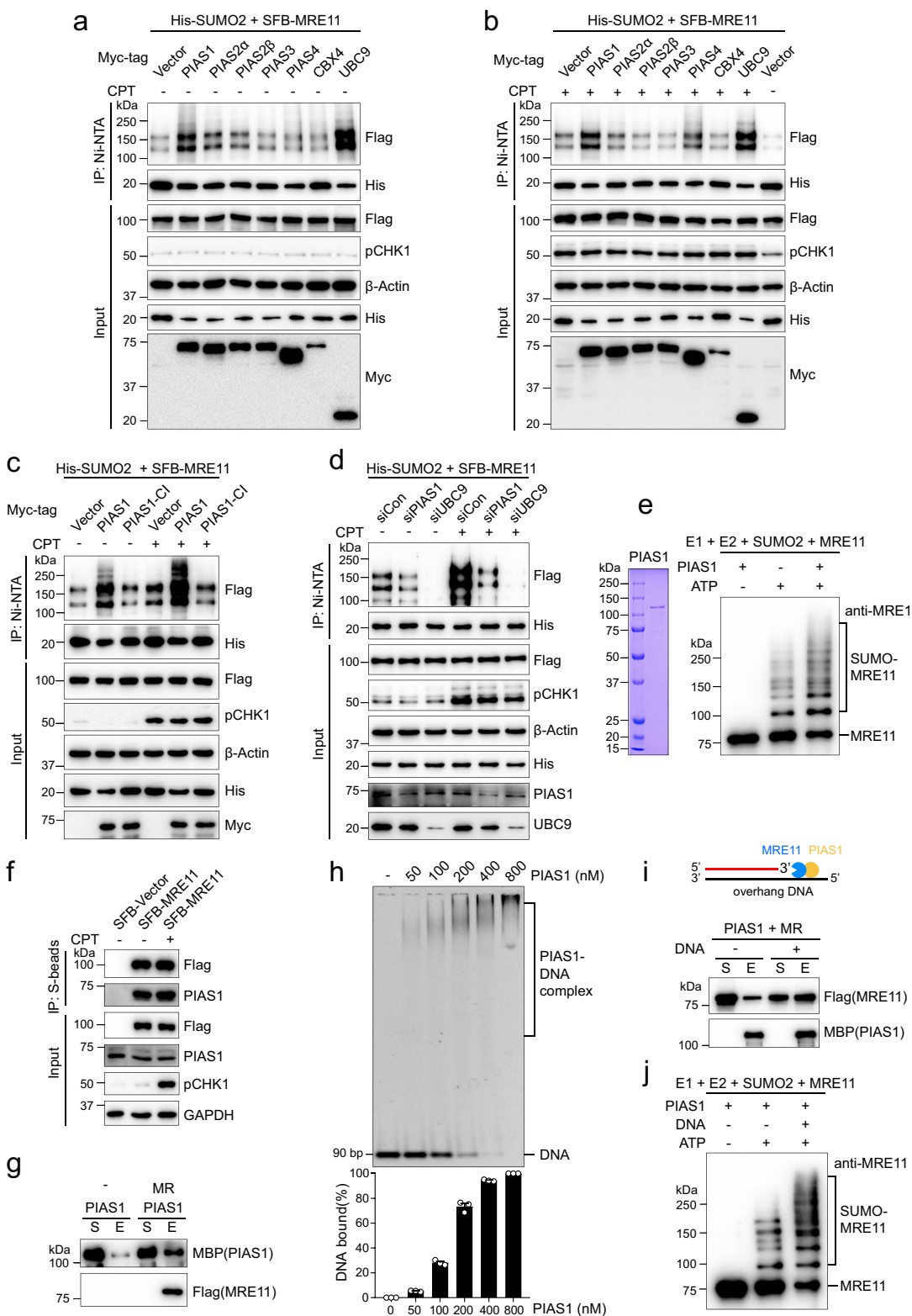

recombinant 4KR fused with a C-terminal 2× or 3× poly-SUMO2 chain (Fig. 6a). We found that the ubiquitylation of 4KR was significantly reduced by the addition of the poly-SUMO2 chain (Fig. 6b). Furthermore, DNA end resection and cell viability following DNA damage were assayed by immunoblotting and cell counting kit-8 (CCK-8) assays. As shown in Fig. 6c, d, the poly-SUMO2 chain fused to 4KR successfully relieved the DNA end resection defect and cell

hypersensitivity to DNA-damaging agents, and the 3×SUMO2 chain is more efficient than the 2×SUMO2 chain. Based on the above observations, we propose that the poly-SUMO2 chain may serve as a shield for MRE11 to block ubiquitin, MRE11 E3 ubiquitin ligases or proteasome-related factors, or recruit certain SIM-containing proteins to prevent MRE11 ubiquitylation, thus enhancing protein stability.

**Fig. 2 | PIAS1 is the major SUMO E3 ligase for MRE11. a**, **b** HEK293T cells transfected with the indicated plasmids were treated with 1 μM CPT for 8 h or not. Then, SUMO-MRE11 was pulled down in guanidine denaturing buffer, and examined by immunoblotting. **c** PIAS1-CI (the catalytically inactive mutant, C346S/C351S/H353A/ C356S) does not enhance MRE11 SUMOylation. **d** MRE11 SUMOylation is abolished by PIAS1 knockdown. **e** Purified PIAS1 protein promotes MRE11 SUMOylation in vitro. MR was incubated with AOS1/UBA2, UBC9, SUMO2, and PIAS1 for 15 min, followed by immunoblotting analysis. **f** Interaction between SFB-MRE11 and endogenous PIAS1 in HEK293T cells was analyzed by co-IP. **g** PIAS1 directly interacts

with MRE11 in vitro. Purified proteins were incubated together and pulled down by anti-Flag beads. The supernatant (S) and eluate (E) fractions were subjected to immunoblotting. **h** PIAS1 binding to 5' overhang DNA in vitro was investigated by electrophoretic mobility shift assay. The data are presented as means ± SEM, $n = 3$ independent experiments. **i** DNA enhances the interaction between MRE11 and PIAS1. MBP-PIAS1 (150 nM) and Flag-MRE11 (50 nM) were incubated in the presence of DNA (90 nM), and their interaction was examined by pull-down assay with amylose beads. **j** PIAS1-catalyzed MRE11 SUMOylation in vitro is further enhanced when DNA is present.

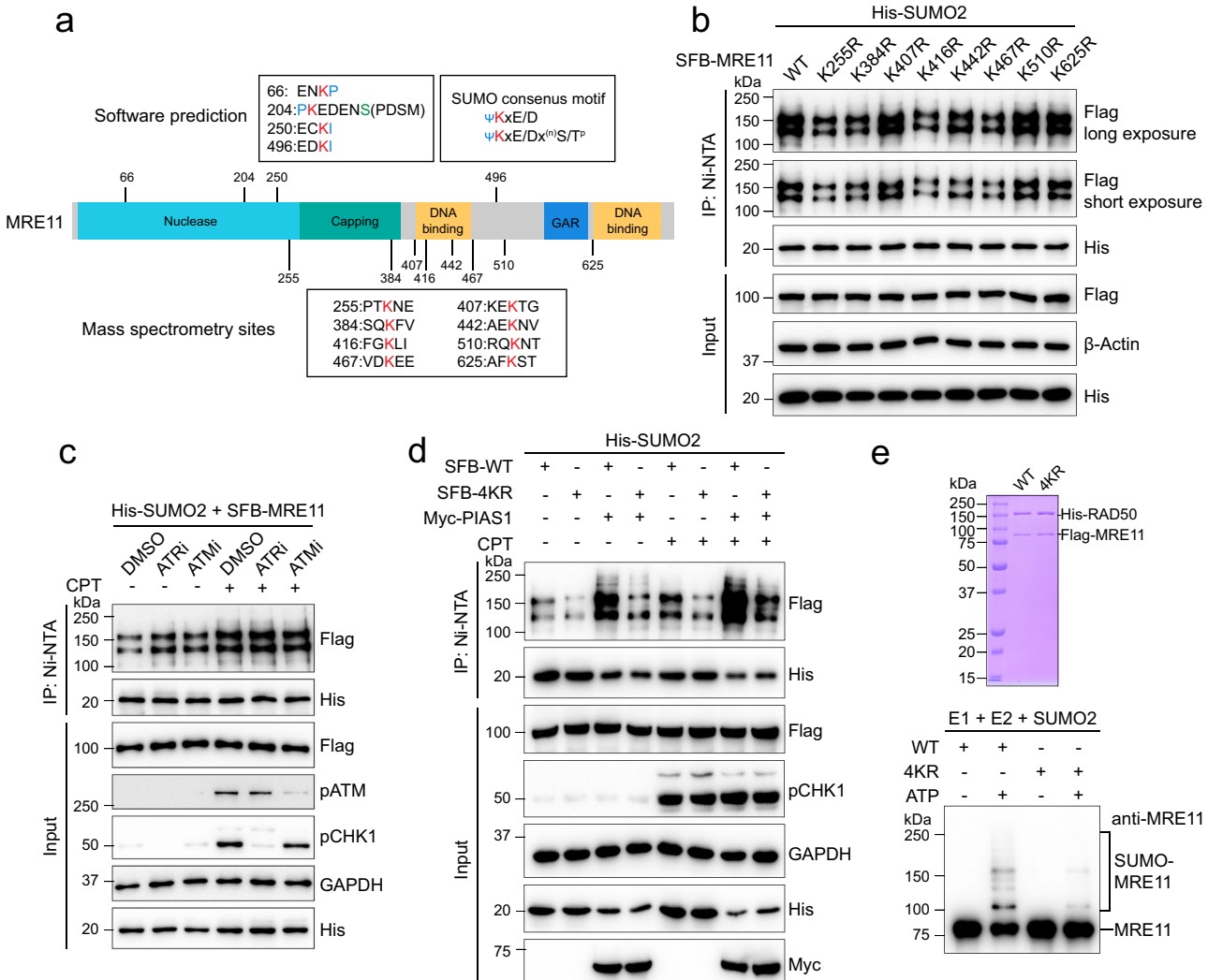

**Fig. 3 | Screening for MRE11 SUMOylation sites. a** Schematic of the SUMOylation sites in MRE11. The predicted high-rank sites analyzed by JASSA, SUMOplot, and GPS-SUMO software, and the mass spectrometry sites from a previous study were listed. **b** K255, K384, K416, and K467 mutations impaired MRE11 SUMOylation. HEK293T cells were transfected with His-SUMO2 and SFB-WT MRE11 or the indicated single point mutants, and then MRE11 SUMOylation levels were analyzed by

denaturing pull-down and immunoblotting. **c** Cells were pretreated with 10 μM ATRi (VE-821) and 10 μM ATMi (KU-55933) for 2 h and further with 1 μM CPT for 8 h, followed by the analysis of MRE11 SUMOylation. **d** SUMOylation of 4KR (containing K255R, K384R, K416R, and K467R) under CPT treatment or PIAS1 overexpression were analyzed. **e** Purified 4KR protein was analyzed by Coomassie blue staining, and then subjected to SUMOylation assay in vitro.

## MRE11 is deSUMOylated by SENP3 to prevent its accumulation on chromatin

SUMOylation is a reversible process, and the deconjugation of SUMO from a substrate is catalyzed by SENPs[39]. To determine which SENP deSUMOylates MRE11, we knocked down each SENP with siRNA. We found MRE11 SUMOylation in SENP3-knockdown cells was markedly increased under both normal and stress conditions (Fig. 7a, b, and Supplementary Fig. 6a). Consistent with this finding, overexpression

of WT SENP3 inhibited MRE11 SUMOylation, while catalytically inactive SENP3 (SENP3-CI, C532S) did not exert a similar effect (Fig. 7c). Meanwhile, we also tested SENP3 effect on MRE11 ubiquitylation and protein half-life, and found MRE11 displayed increased ubiquitylation level and shortened protein half-life in SENP3-overexpressing cells (Supplementary Fig. 6b, c). Moreover, as indicated by co-IP, MRE11 and SENP3 interacted with each other, and their interaction stayed unchanged upon DNA damage or ATMi/ATRi treatment (Fig. 7d, e,

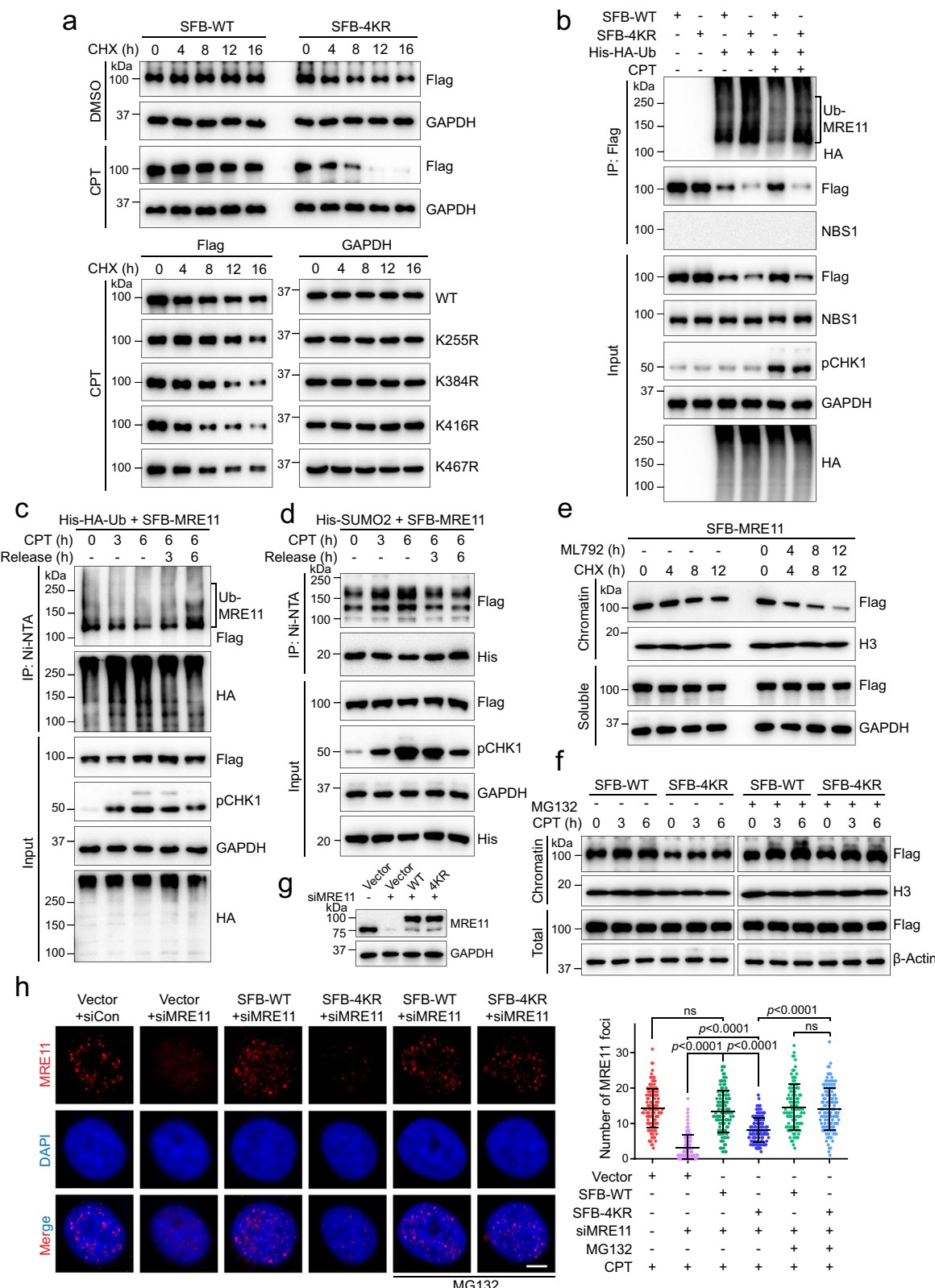

and Supplementary Fig. 6d). Similarly, ATMi and ATRi also did not affect SENP3-catalyzed MRE11 deSUMOylation (Supplementary Fig. 6e). Therefore, the deSUMOylation of MRE11 is mainly catalyzed by SENP3.

To further explore how SENP3 deSUMOylates MRE11, we assessed the localization of SENP3 in response to DNA damage. Immuno-fluorescence assays showed that the majority of SENP3 accumulated in

the cell nucleolus and small amounts were distributed in the nucleo-plasm (Fig. 7f). However, SENP3 migrated from the nucleolus to the nucleoplasm following CPT treatment, and the fluorescence intensity of nucleoplasmic SENP3 sharply decreased after pre-extraction, sug-gesting extranucleolar SENP3 has weak binding to chromatin (Fig. 7f). Interestingly, SENP3 knockdown caused a more obvious increase in MRE11 SUMOylation in the soluble fraction rather than on chromatin

**Fig. 4 | SUMOylation prevents MRE11 from ubiquitin-mediated degradation during DNA end resection. a** HEK293T cells expressing SFB-WT or SFB-4KR were treated with 1 μM CPT for 3 h or not, then washed with fresh medium and followed by 100 μM cycloheximide (CHX) treatment for the indicated times and immunoblotting. **b** HEK293T cells expressing His-HA-Ub, SFB-WT, or SFB-4KR were treated with 1 μM CPT for 8 h or not. The lysates were incubated with anti-Flag beads under SDS denaturing conditions. NBS1 was used as a control. **c** HEK293T cells expressing SFB-MRE11 and His-HA-Ub were treated with 1 μM CPT as indicated, followed by the analysis of MRE11 ubiquitylation. **d** HEK293T cells expressing SFB-MRE11 and His-

SUMO2 were treated with 1 μM CPT as indicated, followed by the analysis of MRE11 SUMOylation. **e** MRE11 on chromatin was degraded faster under the treatment of 20 μM ML792. **f** Recruitment of WT and 4KR to chromatin after 1 μM CPT treatment as indicated was analyzed. **g, h** HeLa cells stably expressing SFB-WT-MRE11 and SFB-4KR with endogenous MRE11 knocked down by siRNA (for 3' UTR) were synchronized in S phase, and MRE11 foci after CPT (1 μM) and MG132 (1 μM) treatment were examined. Scale bar, 5 μm. The data are presented as means ± SD, $n$ (Vector+siCon; Vector+siMRE11; WT + siMRE11; 4KR + siMRE11; WT + siMRE11 + MG132; 4KR + siMRE11 + MG132) = 124; 107; 111; 124; 104; 124 cells, ns = no significance.

(Fig. 7g). Thus, these data suggest that SENP3 deSUMOylates MRE11 primarily in the nucleoplasm instead of on chromatin.

Then, we explored whether SENP3 knockdown influences MRE11 degradation after DNA damage. We compared MRE11 foci numbers in control and siSENP3 cells after 1 h of CPT treatment, and found the number of MRE11 foci was increased in the siSENP3 group (Fig. 7h). However, after a long-term release, the siSENP3 group did not display a declining trend and showed a much higher number of MRE11 foci than the control group (Fig. 7h). Furthermore, SENP3-knockdown cells exhibited excessive DNA end resection and elevated formation of micronuclei after CPT treatment (Fig. 7i and Supplementary Fig. 6f), indicating SENP3 is essential for the maintenance of genome stability.

In summary, the above data suggest that MRE11 needs to undergo deSUMOylation by SENP3 after DNA end resection to prevent its degradation failure and excessive accumulation on chromatin. Failing to deSUMOylate MRE11 results in genome instability due to excessive DNA resection by redundant MRE11.

### MRE11 SUMOylation deficiency is associated with cancer development

MRE11 mutants have been linked to ataxia-telangiectasia-like disorder (ATLD) and are highly correlated with multiple types of cancers[40–42]. According to the cBioPortal and ClinVar databases, many cancer-related mutation sites in MRE11 overlapped with the SUMOylation sites (Fig. 8a). The MRE11 K255E mutation, in particular, has been identified in uterine endometrioid carcinoma (1/399, in cBioPortal database), and both K255E and K384Q/I mutations have been found in hereditary cancer-predisposing syndrome, but their intrinsic effect on cancer development is still unknown. Thus, we constructed MRE11 K255E and K384Q mutants, and assessed their functions. As shown in Fig. 8b, K255E and K384Q mutations caused a decrease in MRE11 SUMOylation. And behaving like 4KR, MRE11 K255E/K384Q mutant also exhibited a short protein half-life (Fig. 8c). Next, we checked DNA end resection and cell viability upon DNA damage. Compared to the control cells, K255E and K384Q mutated cells showed DNA end resection defects after CPT treatment (Fig. 8d) and exhibited more sensitivity to DNA-damaging agents like Olaparib (a PARP inhibitor) and cisplatin (Fig. 8e). Meanwhile, we used reconstituted reactions to test whether these cancer-related mutations affect MRE11 nuclease activity, and found K255E/K384Q protein exhibited a minor nuclease activity impairment compared with WT MRE11 (Fig. 8f). Together, these data indicate that SUMOylation of MRE11 ensures DNA end resection and HR efficiency, which is closely related to genome stability and cancer development.

### Discussion

MRE11 plays an essential role in DNA end resection, controlling the extent of the incisions during DNA damage repair[43,44], and many co-factors have been identified to regulate MRE11 stability and nuclease activity[45]. For example, DYNLL1 suppresses DNA end resection by directly associating with MRE11 in BRCA1-deficient cells[37]. C1QBP interacts with MRE11-RAD50 to form the MRC complex, which stabilizes MRE11 and prevents MRE11 from binding DNA[46]. In addition, MRE11 is also regulated by PTMs like ubiquitylation[36]. MRE11 is ubiquitylated and then removed by UBQLN4 from chromatin after DNA damage[36]. Inactivation of ATPase p97, a pivotal component of the

ubiquitin proteasome degradation system, causes MRE11 to accumulate on damaged chromatin[47]. These studies showed that MRE11 needs to be appropriately degraded after DNA damage repair, especially in cases of excessive resection. However, the mechanism of preventing MRE11 from ubiquitylation-mediated degradation during DNA end resection has not been identified. In this study, we clearly showed that MRE11 SUMOylation dynamically regulated by PIAS1 and SENP3 coordinates with ubiquitylation to control MRE11 homeostasis on chromatin and that SUMOylation is critical for the initiation of DNA end resection (Fig. 9).

For SUMOylation of MRE11, it has been reported that MRE11 is SUMOylated by adenoviral protein E4-ORF3 and then degraded by E1B-55K and E4-ORF6 through the ubiquitin proteasome pathway, which facilitates adenoviral infection[48]. However, another study showed that the protein level of MRE11 remained stable when infected with E4-ORF3 null adenovirus, and MRE11 SUMOylation was still increased after infection[49], implying that this modification by itself may exert antiviral effects. In our study, we further found MRE11 SUMOylation also occurs in response to DNA damage and PIAS1 is the cellular SUMO E3 ligase of MRE11 (Fig. 2a, b). PIAS1 is recruited to damaged chromatin, which facilitates MRE11 SUMOylation to enhance MRE11 stability by antagonizing ubiquitylation. Previous research has shown that knocking down PIAS1 reduced the recruitment of RPA2 to laser-induced damage sites[21], implying that PIAS1 is highly connected with DNA end resection. When DNA end resection is completed, MRE11 needs to be deSUMOylated and degraded because too much MRE11 causes excessive resection, which is deleterious to genome stability[47]. We subsequently found MRE11 was deSUMOylated by SENP3 for its degradation. A recent study has reported genome instability in SENP3-knockout cells[50]. However, the specific functions of SENP3 in DNA damage repair are still unclear. We observed that SENP3 knockdown led to genome instability by causing MRE11 abnormal accumulation on chromatin (Fig. 7h). SENP3 is redistributed from the nucleolus to the nucleoplasm under DNA damage, and SENP3 knockdown increased MRE11 SUMOylation mainly in the nucleoplasm (Fig. 7g), indicating that SENP3 deSUMOylates MRE11 mainly after DNA end resection when MRE11 is released into the nucleoplasm. However, MRE11 was deSUMOylated when SENP3 was overexpressed (Fig. 7c), indicating that SENP3-mediated deSUMOylation may coordinate with PIAS1 to maintain the MRE11 low SUMOylation level on chromatin, and this could prevent MRE11 with excessive SUMOylation from degradation by STUbL. Therefore, how SENP3 deSUMOylates MRE11 during different DNA repair stages remains a mystery, and further exploration is needed.

SUMOylation is critical for orchestrating DNA repair and has been shown to affect protein recruitment, translocation and stability[51–53]. The cellular SUMOylation level is closely monitored by STUbLs. RNF4 is one of the most common STUbLs for many substrates including SUMO1/2/3, SUMO E3 ligases and DDR proteins, which facilitates protein turnover at damaged chromatin[23,54]. However, we did not detect an obvious change in the MRE11 protein level when RNF4 expression was downregulated (Supplementary Fig. 1c). Because STUbLs preferentially recognize substrates with poly-SUMO chains[55,56], we reasoned that MRE11 may be modified mainly by single-SUMO but not poly-SUMO chains. But we further found that the 3×SUMO2 chain fused to MRE11 did not enhance

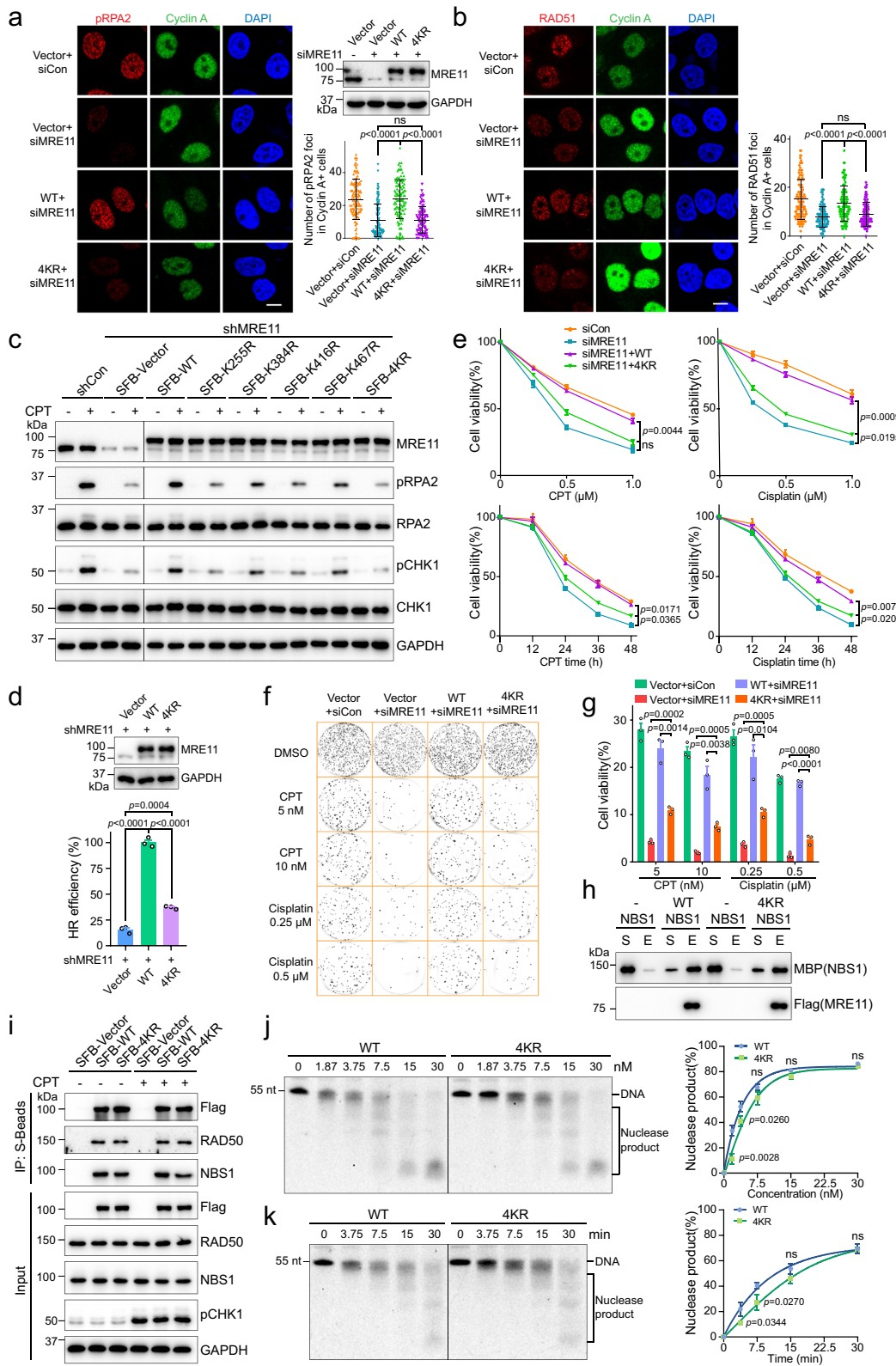

its ubiquitylation and degradation (Fig. 6b), suggesting that MRE11 is not subject to RNF4-mediated degradation.

The crosstalk between phosphorylation and SUMOylation contributes to DNA repair[57]. For example, ATM-mediated CtIP hyperphosphorylation facilitates its SUMOylation and subsequent dissociation at DSB sites[19]. TOPORS phosphorylation by ATM promotes RAD51 SUMOylation upon DNA damage, which is required for RAD51

recruitment and HR efficiency[58]. On the other hand, some SUMOylated sites depend on phosphorylation, known as PDSMs[59]. A previous study showed that MRE11 is phosphorylated by ATM during DNA damage repair, which modulates its DNA-binding activity[46,60]. Here, we further reveal that SUMOylation and deSUMOylation of MRE11 depend on neither ATM- nor ATR-mediated phosphorylation (Fig. 3c and Supplementary Fig. 6e). However, we cannot rule out the involvement of

**Fig. 5 | MRE11 SUMOylation is essential for efficient HR and cell viability.**
**a**, **b** HeLa cells with SUMOylation-deficient MRE11 exhibited impaired pRPA2 (S4/S8) and RAD51 foci formation. Scale bar, 10 μm. The data are presented as means ± SD, *n* (Vector+siCon; Vector+siMRE11; WT + siMRE11; 4KR + siMRE11) = 131; 131; 125; 135 cells for **a**; *n* (Vector+siCon; Vector+siMRE11; WT + siMRE11; 4KR + siMRE11) = 112; 112; 108; 108 cells for **b**, ns = no significance. **c** shMRE11 HeLa cells expressing ectopic SFB-WT and the indicated MRE11 mutants were treated with 1 μM CPT for 1 h or not. Then, cell lysates were subjected to immunoblotting. **d** DR-GFP U2OS cells stably co-expressing SFB-Vector/SFB-WT/SFB-4KR and shMRE11 were infected with I-SceI lentivirus for 48 h, followed by flow cytometric analysis for HR efficiency (means ± SEM, *n* = 3 independent experiments). **e**–**g** The cell viability of stable SFB-WT and SFB-4KR cells with endogenous MRE11 knocked down was detected by CCK-8 assay and colony formation assay (means ± SEM, *n* = 3 independent experiments, ns = no significance). For CCK-8 assay, cells were treated with the indicated concentrations of CPT/cisplatin for 36 h or 1 μM CPT/cisplatin for the indicated times. **h** The assembly of MRN complex containing 4KR MRE11 was assayed by Flag pull-down in vitro. The supernatant (S) containing unbound protein and the eluate (E) were subjected to immunoblotting. **i** The interactions among 4KR MRE11, RAD50 and NBS1 in HEK293T cells were analyzed by co-IP with S-beads. **j** The indicated concentrations of WT and 4KR MRE11 were incubated with DNA for 30 min, and the nuclease products were resolved in denaturing polyacrylamide gels. Quantification of the results is presented (means ± SD, *n* = 3 independent experiments, ns = no significance). **k** Time-course analysis of DNA resection by WT and 4KR MRE11 (10 nM). Electrophoresis and quantification were performed as in **j**.

cyclin-dependent kinases and other kinases in MRE11 SUMOylation. Besides, to the best of our knowledge, NBS1 is the only protein exhibits decreased SUMOylation during the early stage of DNA damage repair, and this phenomenon may be associated with NBS1 phosphorylation (Fig. 1j). Further investigations could try to address whether NBS1 needs to be deSUMOylated before its phosphorylation, and thus promotes DNA end resection and HR.

Previous studies have revealed that dynamic SUMOylation is associated with the occurrence and development of various diseases[31], for example, SUMOylation of the pro-oncogene Akt increases its kinase activity and regulates cell proliferation[61]. Mutations in DJ-1 have been shown in familial Parkinson's disease, and SUMOylation-deficient DJ-1 aggregation causes mitochondrial dysfunction[62,63]. In our study, we also explored the connections between MRE11 SUMOylation and cancers. We found that cancer-related MRE11 K255E and K384Q mutants were prone to degradation after treatment with DSB-inducing agents, which led to failed DNA end resection and thus may result in genome instability and tumorigenesis. In addition, MRE11 SUMOylation-deficient cells were sensitive to chemotherapeutic agents such as PARP inhibitors (Fig. 8e). Because ML792 triggered HR defects, the combination treatment of SUMO inhibitors and PARP inhibitors may exert a synergistic antitumor effect. In summary, we provide evidence supporting that dynamic SUMOylation of MRE11 functions in concert with ubiquitylation to promote DNA end resection and the maintenance of genome stability.

# Methods

## Cell culture
Human HEK293T and HeLa cells were cultured in Dulbecco's modified Eagle's medium (DMEM, Gibco) supplemented with 10% fetal bovine serum (Yeasen) at 37 °C under 5% $CO_2$. The cells were passaged at approximately 90% confluence by using 0.25% trypsin-EDTA. The chemicals used in this study to treat cells are listed in Supplementary Table 1.

## Construction of plasmids and siRNA
The coding sequences of MRE11, NBS1, PIAS1, SENP3, SUMO1, SUMO2, and SUMO3 were cloned into the pMH-SFB vector or pcDNA3.1 fused with different tags. Point and deletion mutations were generated by PCR-based mutagenesis assay and verified by sequencing. Transient plasmid transfection was performed by using Lipofectamine 2000 (Invitrogen). The transfection reagent Lipofectamine RNAiMAX (Invitrogen) was used for siRNA transfection. The siRNA (RuiBiotech) sequences are summarized in Supplementary Table 2.

## Generation of shRNA cell lines and stable expression cell lines
Endogenous MRE11-knockdown cell lines were produced by lentivirus infection, which was generated by cotransfection of the lentiviral vector pLKO.1-shMRE11, envelope plasmids pMD2.G and psPAX2 in HEK293T cells. Then, transfected cells were selected and cultured in DMEM supplemented with 10% fetal bovine serum and puromycin (2 mg/L). The shMRE11 sequence (for 3′ UTR) is 5′-GAGCAUAACUCCAUAAGUA-3′. Stable expression cell lines were generated by transfection with SFB-MRE11 or its point mutants, and then screened by puromycin as above. Monoclonal cells were picked and identified by immunoblotting.

## Quantitative PCR
Total RNA was extracted from cells using TRIzol reagent (Invitrogen) according to the manufacturer's instructions, and cDNA was synthesized by reverse transcription-PCR. qRT-PCR was performed using SYBR Premix on an ABI Detector (StepOne Plus). GAPDH was used as a control gene. Each group was performed in triplicate. The relative expression level of mRNA was calculated by the $2^{-\Delta\Delta Ct}$ method. The primers (RuiBiotech) are listed in Supplementary Table 3.

## Immunoblotting
Cells (1 × 10⁶) were lysed in 300 μL high salt NETN buffer (20 mM Tris-HCl, pH 8.0, 400 mM NaCl, 1 mM EDTA, 0.5% NP40) supplemented with 1 mM PMSF, 20 mM NEM, and 1 μg/mL aprotinin, and the cell lysates were clarified by centrifugation (1.8 × 10⁴ g for 15 min at 4 °C). Protein samples were resolved by SDS-PAGE and then transferred to PVDF membranes. After blocking with 5% BSA in Tris-buffered saline containing Tween (TBST) for 1 h, membranes were incubated with different primary antibodies, followed by HRP-conjugated secondary antibodies. The signal was detected using an ECL immunoblotting kit. All the uncropped and unprocessed scans of blots in this study can be found in the Source Data.

Antibodies used in immunoblotting were as follows: anti-Flag-HRP (Sigma-Aldrich, A8592, 1:2000); anti-MBP-HRP (NEB, E8038, 1:3000); Rabbit-anti-MRE11 (Proteintech, 10744-1-AP, 1:1000); Rabbit-anti-NBS1 (Proteintech, 55025-1-AP, 1:1000); Rabbit-anti-RAD50 (ABclonal, A3078, 1:1000); Rabbit-anti-RNF4 (Proteintech, 17810-1-AP, 1:1000); Rabbit-anti-UBC9 (Proteintech, 51018-2-AP, 1:2000); Rabbit-anti-PIAS1 (Proteintech, 23395-1-AP, 1:1000); Rabbit-anti-phospho-RPA32 (S4/S8) (Abcam, ab87277, 1:1000); Rabbit-anti-RPA32 (Proteintech, 10412-1-AP, 1:1000); Rabbit-anti-phospho-CHK1 (S345) (Cell Signaling Technology, 2348 S, 1:1000); Mouse-anti-CHK1 (Santa Cruz, sc-8408, 1:1000); Rabbit-anti-phospho-H2AX (S139) (Cell Signaling Technology, 9718, 1:1000); Rabbit-anti-phospho-ATM (S1981) (Cell Signaling Technology, 5883 S, 1:1000); Rabbit-anti-phospho-NBS1 (S343) (Cell Signaling Technology, 3001 S, 1:1000); Rabbit-anti-SUMO2/3 (Santa Cruz, sc32873, 1:500); Mouse-anti-SUMO2/3 (Abcam, ab81371, 1:500); Mouse-anti-SUMO1 (Santa Cruz, sc-5308, 1:200); Mouse-anti-Histone H3 (Biodragon, B1055, 1:2000); Mouse-anti-Myc (Santa Cruz, sc-40, 1:1000); Rabbit-anti-HA (BioLegend, 901503, 1:1000); Rabbit-anti-His (Santa Cruz, sc-803, 1:1000); Mouse-anti-GAPDH (Sungene, KM9002, 1:5000); Mouse-anti-β-Actin (Sungene, KM9001, 1:5000).

## Immunofluorescence
HeLa cells were seeded on coverslips and treated with 1 μM CPT for the corresponding times. Cells were then fixed in cold 4% paraformaldehyde for 15 min at 4 °C and permeabilized in 0.25% Triton X-100

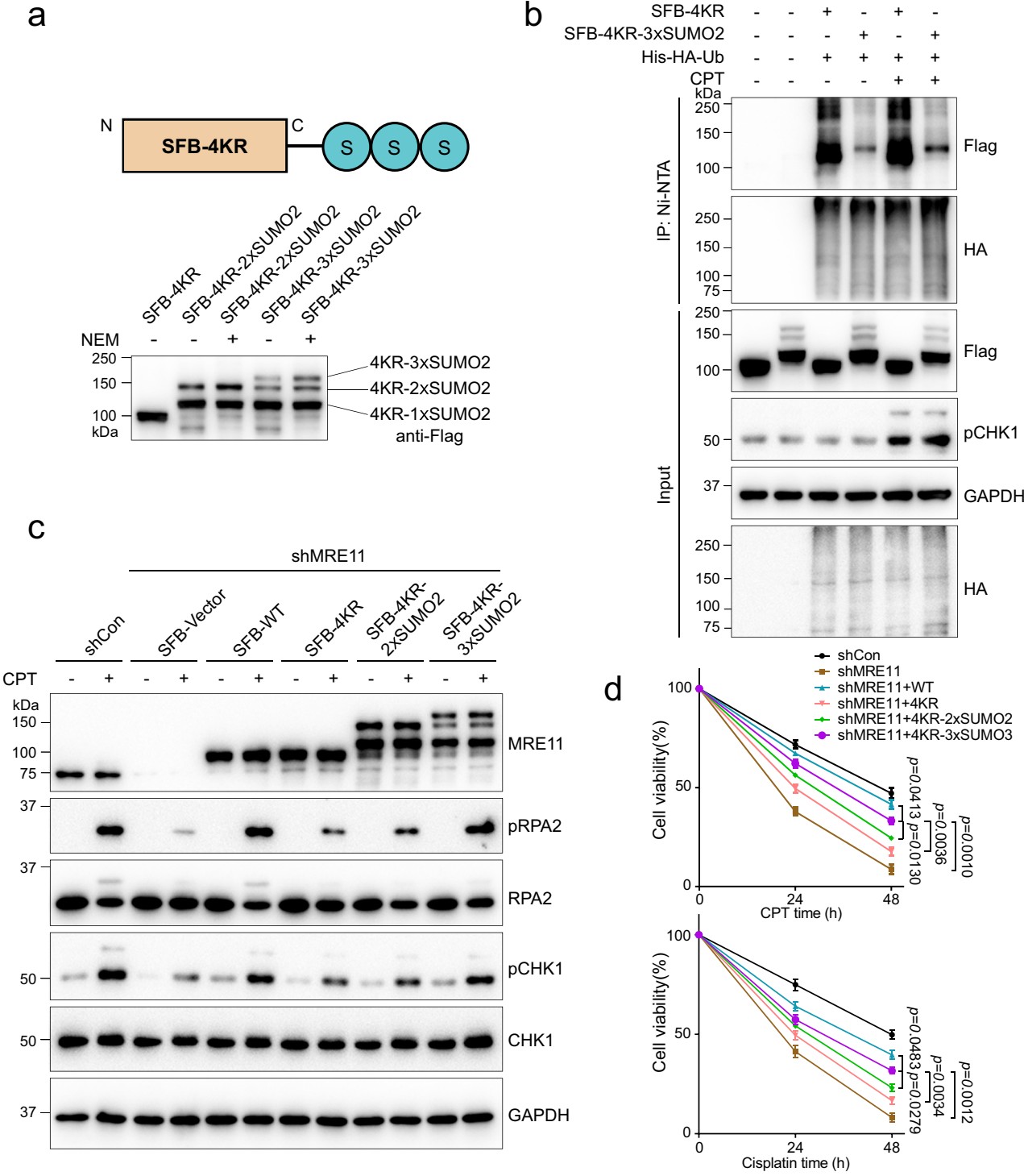

**Fig. 6 | The stability of SUMOylation-deficient MRE11 is improved by fusion with a poly-SUMO2 chain. a** Schematic of the 4KR-MRE11 protein with a 3×SUMO2 chain fused to the C-terminus. Fusion protein expression was detected by immunoblotting. **b** The ubiquitylation of 4KR was inhibited by fusion with the 3×SUMO2 chain. **c** SUMOylation-deficient MRE11 fused with poly-SUMO2 facilitated RPA2 phosphorylation and ATR-CHK1 pathway activation after DNA damage. shMRE11 HeLa cells reconstituted with SFB-WT, SFB-4KR, and SFB-4KR-poly-SUMO2 as indicated were treated with 1 μM CPT for 1 h or not. **d** Cell viability was determined by CCK-8 assay after treatment with 0.5 μM CPT or 1 μM cisplatin for the indicated times (means ± SEM, $n = 3$ independent experiments).

solution for 5 min at room temperature. For MRE11 staining, pre-extraction was first performed with 0.5% Triton X-100 solution for 10 min to release MRE11 in the soluble fraction, followed by fixation with 4% paraformaldehyde for 20 min. After blocking with 2% BSA for 10 min, the primary antibodies were added into the cells, and then incubated for 1 h at room temperature, followed by secondary

antibody incubation for 30 min. Finally, the cells were stained with DAPI for 5 min, and images were acquired using ZEN software or Leica LAS X software.

Antibodies used in immunofluorescence were as follows: Rabbit-anti-phospho-RPA32 (S4/S8) (Abcam, ab87277, 1:500); Rabbit-anti-RAD51 (Abcam, ab133534, 1:250); Mouse-anti-Cyclin A (Santa Cruz,

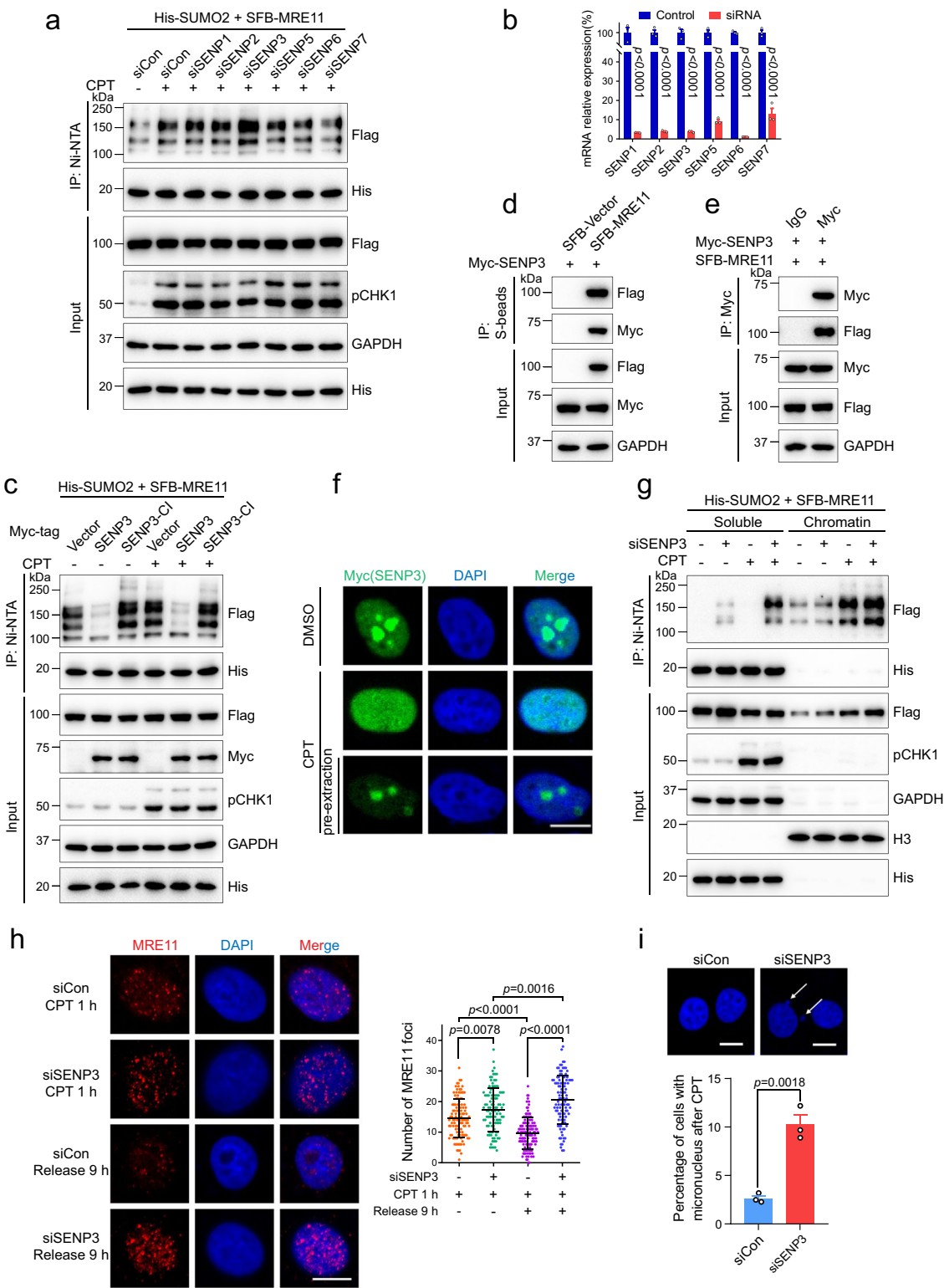

sc-271682, 1:250); Rabbit-anti-MRE11 (Abcam, ab33125, 1:200); Mouse-anti-Myc (Santa Cruz, sc-40, 1:250).

**Soluble fractions and chromatin fractions extraction**
Cells ($2 \times 10^6$) were gently lysed with 400 μL EBC A buffer (50 mM Tris-HCl, pH 7.5, 100 mM NaCl, 0.05% NP40, 1 mM EDTA, 1 mM DTT, 1 mM PMSF, 2 μg/mL aprotinin, 20 mM NEM) for 8 min. After centrifugation (800 g for 5 min at 4 °C), the supernatants were soluble fractions. The remaining parts were washed with EBC A buffer 2 times and cold PBS 1 time and subsequently extracted with 120 μL EBC B buffer (50 mM Tris-HCl, pH 7.5, 0.5 mM EDTA, 300 mM NaCl, 5 mM CaCl$_2$, 10 U micrococcal nuclease, 1 mM PMSF, 2 μg/mL aprotinin, 20 mM NEM) for 10 min at room temperature. After sonication and centrifugation ($1.8 \times 10^4$ g for 15 min at 4 °C), the supernatants were chromatin

**Fig. 7 | SENP3 deSUMOylates MRE11 mainly after DNA end resection. a** HeLa cells were cotransfected with SFB-MRE11, His-SUMO2, and the indicated siRNA, followed by the treatment of 1 μM CPT for 8 h. MRE11 SUMOylation was analyzed by Ni-NTA pull-down and immunoblotting. **b** The knockdown efficiency of SENPs as in **a** was detected by quantitative real-time PCR (means ± SEM, $n = 3$ independent samples). **c** MRE11 SUMOylation in HEK293T cells expressing Myc-tagged SENP3 or SENP3-CI (the catalytically inactive mutant, C532S) was analyzed as in **a**. **d, e** The mutual interaction between MRE11 and SENP3 was examined by co-IP. **f** HeLa cells expressing Myc-tagged SENP3 were treated with 1 μM CPT for 1 h, and then subjected to immunofluorescence assay with or without pre-extraction. Scale bar, 10 μm. **g** SENP3-knockdown HeLa cells were transfected with SFB-MRE11 and His-SUMO2. After the treatment with 1 μM CPT for 6 h, soluble and chromatin fractions were isolated, and MRE11 SUMOylation was examined by Ni-NTA pull-down and immunoblotting. **h** SENP3-knockdown HeLa cells were treated with 1 μM CPT and released as indicated times. Then, MRE11 foci were analyzed. Scale bar, 10 μm. The data are presented as means ± SD, $n$ (siCon+CPT 1 h; siSENP3+CPT 1 h; siCon +Release 9 h; siSENP3+Release 9 h) = 115; 104; 120; 107 cells. **i** SENP3-knockdown cells were treated with 1 μM CPT for 1 h and released for 9 h. DAPI staining was performed, and micronuclei were counted in over 500 cells. Data are presented as means ± SEM, $n = 3$ independent experiments. Scale bar, 10 μm.

fractions. The soluble and chromatin fractions were analyzed by SDS-PAGE and immunoblotting.

### In vivo SUMOylation and ubiquitylation denaturing pull-down assay

To test the SUMOylation and ubiquitylation of MRE11, we used two methods with different denaturing buffers.

(1) SDS denaturation method: Cells in 6 cm dishes were lysed in 300 μL SDS-A buffer (100 mM Tris-HCl, pH 6.8, 1% SDS, 10% glycerol) freshly supplemented with 1 mM PMSF, 2 μg/mL aprotinin and 20 mM NEM, boiled for 10 min and then 7× diluted with SDS-B buffer (100 mM Tris-HCl, pH 6.8, 10% glycerol, fresh 1 mM PMSF, 2 μg/mL aprotinin and 20 mM NEM). Then, the samples were sonicated and centrifuged ($1.8 \times 10^4$ g, 15 min, 4 °C) to remove debris. The lysates were immunoprecipitated with MRE11 antibody and Protein A beads (Thermo Scientific, Cat#20333) for endogenous MRE11 or anti-Flag M2 agarose beads for SFB-MRE11. Then, the beads were washed with SDS-B buffer 3 times and eluted in 2× SDS loading buffer.

(2) Guanidine/urea denaturation method: Cells in 6 cm dishes were transfected with 10×His SUMO1/2 or 6×His-HA-ubiquitin. After 48 h, the cells were lysed in 1 mL buffer I (6 M guanidine-HCl, 0.1 M $Na_2HPO_4$, 6.8 mM $NaH_2PO_4$, 10 mM Tris-HCl, pH 7.5, 0.1% Triton X-100, 100 mM NaCl, freshly added 10 mM β-mercaptoethanol, 20 mM imidazole). After sonication and centrifugation, the supernatant was incubated with HisSep Ni-NTA Agarose Resin (Yeasen, Cat#20503ES10) on a rotator for 3 h at room temperature. Then, the beads were successively washed with 1 mL buffer II (8 M urea, 0.1 M $Na_2HPO_4$, 6.8 mM $NaH_2PO_4$, 10 mM Tris-HCl, pH 7.5, 0.1% TritonX-100, 100 mM NaCl, freshly added β-mercaptoethanol, 20 mM imidazole) 1 time and 1 mL Buffer III (8 M urea, 18 mM $Na_2HPO_4$, 80 mM $NaH_2PO_4$, 10 mM Tris-HCl, pH 6.3, 0.1% TritonX-100, 100 mM NaCl, freshly added 10 mM β-mercaptoethanol, 20 mM imidazole) 3 times, and eluted in 50 μL Buffer IV (400 mM imidazole, 0.15 M Tris-HCl, pH 6.7, 150 mM NaCl, 20% glycerol, 100 mM β-mercaptoethanol, 3.3% SDS). Immunoblotting was performed as above.

### Expression and purification of the MRE11/RAD50 complex and NBS1

The MRE11/RAD50 complex and NBS1 were overexpressed in yeast and High Five cells, respectively, and purified to near homogeneity using affinity purification. Specifically, the pESC-URA vector containing Flag-tagged MRE11 and His-tagged RAD50 was introduced into the protease-deficient yeast strain. After 24 h of culture, yeast cells expressing MRE11/RAD50 were harvested and lysed. The clarified cell lysate was first incubated with Ni-NTA agarose resin in T buffer containing 300 mM KCl, protease inhibitors, and 15 mM imidazole for 2–3 h at 4 °C, and then the beads were washed with T buffer containing 300 mM KCl and 15 mM imidazole 4 times. The proteins eluted from Ni-NTA agarose resin were further purified with anti-Flag M2 affinity beads (Sigma-Aldrich, Cat#A2220). For NBS1 expression, the pFastBac vector containing MBP- and His-tagged NBS1 was used, bacmid and baculovirus were prepared in DH10Bac and Sf9 cells, respectively. High

Five cells infected with high-titer P3 baculovirus were harvested and lysed in T buffer containing 300 mM KCl. The clarified cell lysate was first incubated with amylose agarose beads (NEB, Cat#E8035S) in T buffer containing 300 mM KCl and protease inhibitors for 2-3 h at 4 °C, then the beads were washed with the same buffer 4 times. The proteins eluted from beads were further subjected to affinity purification with Ni-NTA agarose resin as above. The purified MRE11/RAD50 and NBS1 proteins were stored at −80 °C in small aliquots.

### Expression and purification of SUMO1/2, AOS1/UBA2, UBC9 and PIAS1

The His-tagged recombinant protein expression vectors pET-SUMO1/2, pET-AOS1/UBA2 and pET-UBC9, and the His- and MBP-tagged pET-PIAS1 expression vectors were constructed. These vectors were transformed into the BL21 (DE3) *E. coli* strain, and 0.1 mM IPTG was added to induce protein expression. After 14 h at 16 °C, bacteria were harvested and lysed. The lysates were incubated with Ni-NTA agarose resins in T buffer containing 150 mM KCl, protease inhibitors, and 20 mM imidazole for 2–3 h at 4 °C, and then the beads were washed with T buffer containing 150 mM KCl 4 times. Then, the proteins were eluted with T buffer containing 150 mM KCl and 200 mM imidazole. For PIAS1 purification, further incubation with amylose agarose beads was performed at 4 °C for 4 h. After washing 3 times with T buffer containing 150 mM KCl, PIAS1 was eluted with maltose solution, and all proteins were stored at −80 °C.

### In vitro SUMO conjugation assay

A total of 280 ng Flag-MRE11 (complexed with RAD50), 900 ng SUMO1/2, 280 ng AOS1/UBA2, 280 ng UBC9, and 280 ng PIAS1 were added into an in vitro SUMOylation reaction with 5× reaction buffer (25 mM Tris-HCl, pH 7.5, 5 mM MgCl$_2$, 50 mM KCl, 2 mM ATP), and each reaction was supplemented with ddH$_2$O to 24 μL. The reaction mixture was incubated at 37 °C for 10–45 min as indicated in different experiments, and the reaction was stopped by adding 4× SDS loading buffer. The reaction products were detected by SDS-PAGE and immunoblotting.

### Electrophoretic mobility shift assay (EMSA)

The indicated concentrations of PIAS1 were mixed with 5' overhang DNA substrate in 12 μL of reaction buffer (25 mM Tris-HCl, pH 7.5, 2 mM MgCl$_2$, 1 mM DTT, 100 μg/mL BSA) containing 50 mM KCl. After 20 min incubation at 37 °C, the reactions were mixed with 4 μL of 4× loading buffer (20 mM Tris-HCl, pH 7.5, 40% glycerol, 2 mM EDTA, 0.2% orange G) before electrophoresis with 4% native polyacrylamide gels in 0.5× TBE buffer (2 mM EDTA, 90 mM Tris-HCl, 90 mM boric acid, pH 8.3). After electrophoresis, the images were acquired by Genesys and then subjected to ImageJ analysis.

### In vitro pull-down assay

To evaluate the interaction between SUMOylation-deficient MRE11 (4KR), WT-MRE11 and PIAS1 or NBS1. Purified Flag-WT-MRE11 (complexed with RAD50) and Flag-4KR (complexed with RAD50) were incubated with MBP-PIAS1 or MBP-NBS1 in T50 buffer (25 mM Tris-HCl,

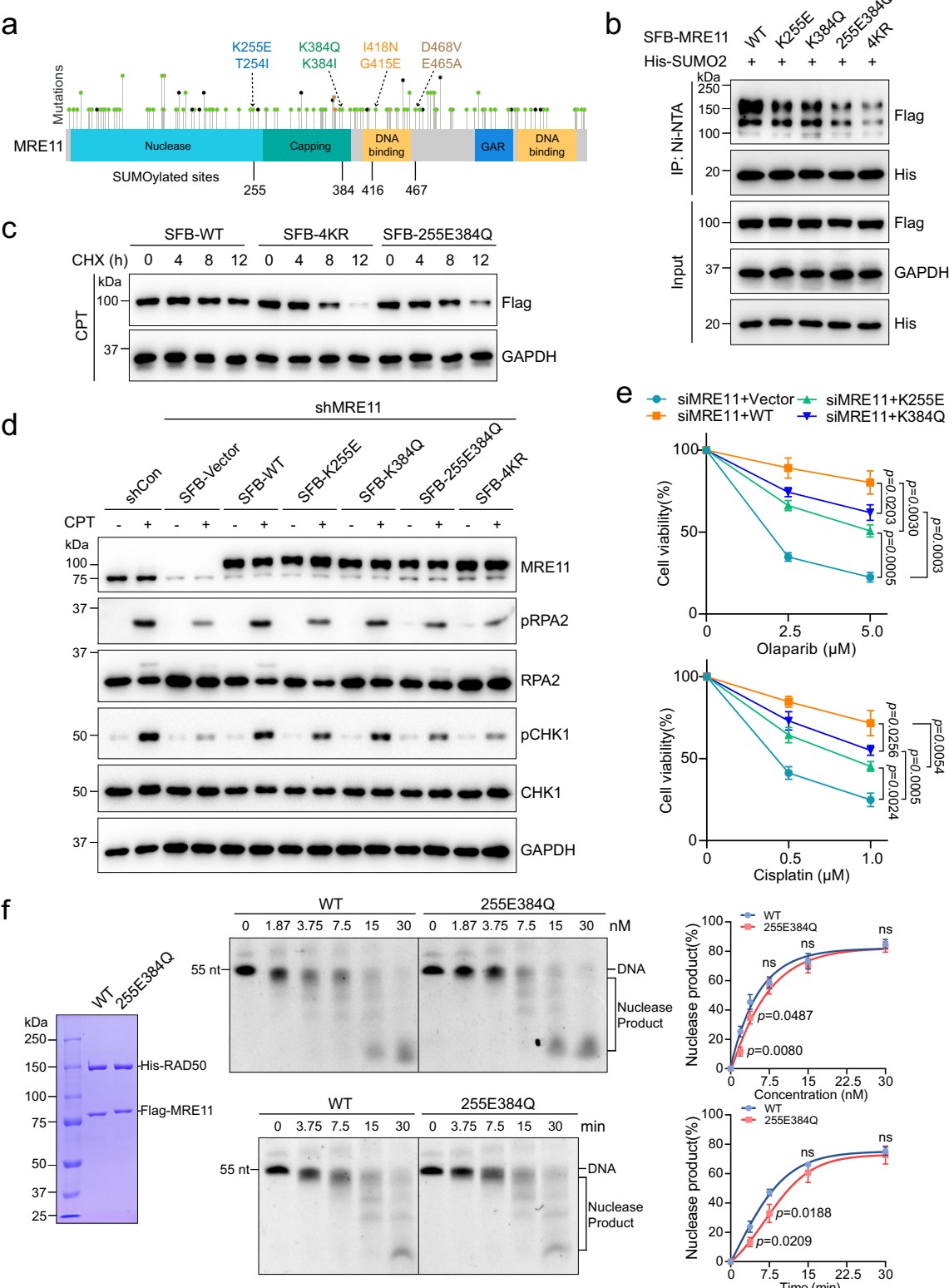

**Fig. 8 | MRE11 SUMOylation defects relate to cancer development. a** Overlap between MRE11 SUMOylation sites and disease-related mutation sites. MRE11 mutation sites were collected from the ClinVar and cBioPortal databases. **b** The cancer-related MRE11 mutants showed a decrease in SUMOylation. **c** 255E384Q mutant was easier to degrade after the treatment of CPT (1 μM, 4 h). **d** HeLa cells expressing the cancer-related MRE11 mutants exhibited defects in RPA2 phosphorylation and the ATR-CHK1 pathway activation upon CPT treatment. **e** HeLa cells with K255E or K384Q mutant were sensitive to olaparib and cisplatin at the indicated doses (means ± SEM, $n = 3$ independent experiments). **f** Purified 255E384Q protein was analyzed by Coomassie blue staining. The nuclease assay in vitro was performed as in Fig. 5j, k (means ± SD, $n = 3$ independent experiments, ns = no significance).

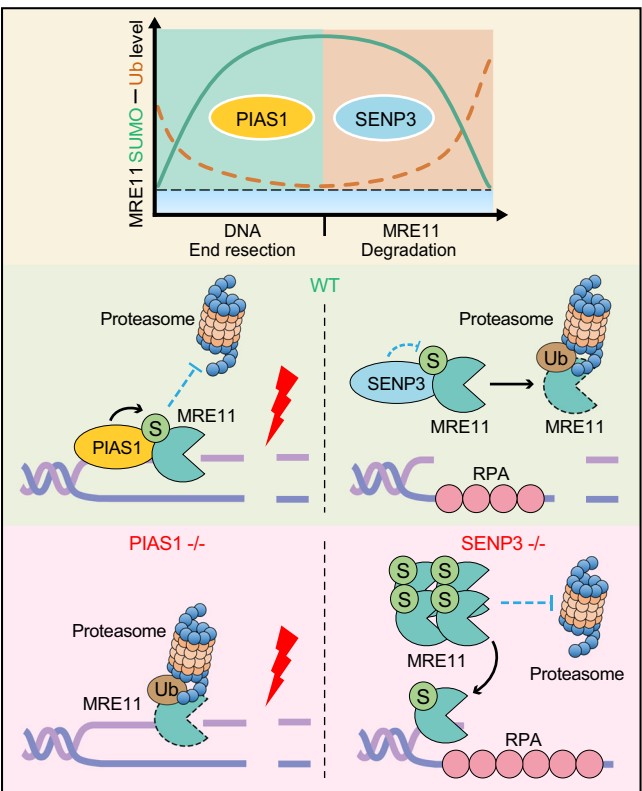

**Fig. 9 | A working model of the MRE11 SUMOylation in DNA damage repair.** The PIAS1-SENP3 axis dynamically controls the SUMOylation and ubiquitylation levels of MRE11 to facilitate DNA end resection. Upon DSB induction, the SUMO E3 ligase PIAS1 and MRE11 are recruited to broken DNA ends, and MRE11 SUMOylation is enhanced to antagonize ubiquitin-mediated proteasomal degradation during DNA end resection. Mainly after DNA end resection, SENP3 deSUMOylates MRE11 and prevents the excessive accumulation of MRE11.

pH 7.5, 10% glycerol, 0.5 mM EDTA, 50 mM KCl, 1 mM DTT, 0.01% Igepal) on ice for 2 h. Then, 15 μL of anti-Flag M2 affinity beads or amylose beads were added to the solution mixture to pull down the target protein. After an additional 3 h incubation, the supernatants were collected by centrifugation, and then the beads were washed with T50 buffer 4 times and eluted in 2× SDS buffer for 5 min at 95 °C. The collected samples were analyzed by immunoblotting.

### Nuclease reactions
To detect SUMOylation-deficient MRE11 and WT-MRE11 nuclease activities, the indicated concentrations of 4KR (complexed with RAD50) or WT-MRE11 (complexed with RAD50) were mixed with 10 nM 5′ overhang DNA in reaction buffer (25 mM Tris-HCl, pH 7.5, 2 mM $MnCl_2$, 1 mM DTT, 10 μg/mL BSA, 2 mM ATP) containing 100 mM KCl, followed by incubation for the indicated times at 37 °C. The nuclease products were loaded onto 12% urea polyacrylamide gels in TAE buffer (40 mM Tris-HCl, pH 8.3, 1 mM EDTA) and then subjected to imaging analysis (ImageJ).

### HR assay
DR-GFP reporter U2OS cell line was gifted from Prof. Jiadong Wang (Peking University). DR-GFP U2OS cells stably co-expressing shMRE11 and SFB-Vector/SFB-WT-MRE11/SFB-4KR were infected with I-SceI lentivirus. After 48 h infection, cells were collected and subjected to flow cytometric analysis to detect GFP-positive cells. The gating strategy was shown in Supplementary Fig. 5a. HR efficiency was calculated as the fold change of the percentage of GFP-positive cells normalized to the WT group. The data were subjected to FlowJo analysis.

### Cell survival assay
Cell survival assay was performed using CCK-8 kits (Donjindo, Cat#CK04) according to the manufacturer's protocols. Cells ($1 \times 10^3$ cells/well) were seeded in 96-well plates. After the indicated times, 10 μL CCK-8 solution was added to each well, and the cells were incubated for 2 h. DMEM containing 10% CCK-8 was used as the control. The absorbance at 450 nm was detected by a microplate reader. The OD was calculated according to the formula: (ODexperiment − ODblank) − (ODcontrol − ODblank). Cell survival was calculated as the fold change of the OD normalized to the untreated drug group.

### Colony formation assay
HeLa cells stably overexpressing WT-MRE11 or 4KR were transfected with siMRE11 (for 3′ UTR). After 36 h, the cells were seeded into 6 cm dishes at a density of $8 \times 10^2$ per dish. Then, the cells were treated with DNA-damaging drugs as indicated for 24 h, and the medium was replaced. Two weeks after drug treatments, cells were stained with 0.1% crystal violet dyes for 40 min. The dishes were washed with water and the cell colonies were counted.

### Statistics and reproducibility
Each experiment was repeated at least three times as indicated in the figure legend. Two-sided Mann-Whitney-Wilcoxon rank sum tests were employed to identify significance for foci statistical results by SPSS 17.0. The other experiments were performed using Two-tailed Student's t-tests with GraphPad Prism 8. $p < 0.05$ was considered to indicate significant differences, and the statistical details are shown in the figures and figure legends.

### Reporting summary
Further information on research design is available in the Nature Research Reporting Summary linked to this article.

## Data availability
All data supporting the findings of this study are available within the manuscript file and its Supplementary Information files. ClinVar (https://www.ncbi.nlm.nih.gov/clinvar) and cBioPortal (https://www.cbioportal.org) databases are publicly available. Source data are provided with this paper.

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

## Acknowledgements

We thank all members of the laboratory for insightful discussion and technical assistance. We thank Dr. Shiwei Li (Peking University) for technical assistance of DR-GFP assay. This work was supported by grants from the National Natural Science Foundation of China (81972608 and 82172951 to W.W.).

## Author contributions

T.Z. and H.Y. performed the experiments. Z.Z., Y.B. provided technical and experimental assistance. T.Z. and W.W. designed the experiments and wrote the manuscript. J.W. and W.W. proofread and revised this manuscript.

## Competing interests

The authors declare no competing interests.
