## [Peer Review File · Nature Communications]

Crosstalk between SUMOylation and ubiquitylation controls DNA end resection by maintaining MRE11 homeostasis on chromatinREVIEWER COMMENTS

Reviewer #1 (Remarks to the Author):

The manuscript from Zhang and colleagues characterized a novel post-translational regulatory mechanism of Mre11 upon DNA damage. They showed that Mre11 is SUMOylated upon DSB induction and is mediated by the SUMO E3 ligase PIAS1. Moreover, they found that SUMOylation of Mre11 is able to prevent it from ubiquitin-mediated degradation during DSB resection. In cells, expression of SUMOylation defective Mre11 led to defective-end resection and compromised homologous recombination. Overall, the data are convincing and are largely consistent with the authors' model. It should be well accepted by the genome research field. Below are a few additional comments that can help the authors to clarify and strength this already nicely performed work.

1. Virtually all evidence for Mre11 SUMO modification was gathered while overexpressing His-tagged or Flag-tagged SUMO. Do the authors have evidence of DSB-induced modification of endogenous Mre11 by endogenous SUMO?
2. Can the interaction between Mre11 and PIAS1/SEN3 be regulated by DNA damage-inducing agents such as CPT and etoposide?
3. The authors suggest that SUMOylation of Mre11 prevents it from ubiquitin-mediated degradation when resecting damaged chromatin. The half-life of wild-type Mre11 and its 4KR mutant should be determined in the presence/absence of PIAS1/SEN3.
4. The authors should use the well-established HR reporter assay (measured by the DR-GFP) to determine the impact of Mre11 SUMOylation on homologous recombination repair.

Reviewer #2 (Remarks to the Author):

In this manuscript, the authors describe a dynamic post-translational modification of the nuclease MRE11 in response to DNA damage. Not only they are able to describe this phenomenon, but also its relevance for DNA resection and survival to genotoxic agents and the SUMOylated residues of the protein. Furthermore, they are able to describe both the SUMO E3 ligase (PIAS1) and the SUMO deconjugated enzyme (SEN3) involved in the process. Finally, they can propose a mechanism in which transient sumoylation of MRE11 prevents its ubiquitination, protecting the protein from degradation during a temporal window to allow resection. Later, SEN3-mediated SUMO deconjugation facilitates MRE11 removal in order to avoid hyper-resection. The relevance of this process is shown as mutations that impair this SUMOylation are found in cancer samples. Overall, the authors present a compelling story that is of interest of the field. In general, the data strongly support the conclusions of the authors, results are well presented and the manuscript is a solid candidate for publication in Nat Comm. It is true that opens interesting questions, such as how the temporal dynamic SUMOylation/deSUMOylation upon DNA damage is achieved as it looks ATM/ATR independent. Although it looks as it relies on a change on the nuclear distribution of SEN3, how this is achieved is not clear. In any case, in my opinion the advance this manuscript presents is enough at this stage to grant publication, leaving those open questions for further follow up analysis.

Having said that, I have a grave concern about the in vivo analysis of resection that has to be clarified in order to get my full support for its publication. RPA foci is analyzed in S/G2 cells marked as Cyclin A positive upon CPT treatment. I have two important issues to mention:

1. Most Cyclin A positive cells will be in G2, especially those that show a strong staining. But CPT will only create breaks in S phase cells. Thus, most cyclin-positive cells should not show RPA foci. From the data presented it does not look like that. Either the text is misleading or there is something strange on the experiment and G2 cells are indeed suffering chromosome breaks.
2. Technically, I do not understand how the authors can visualize RPA foci without pre-extraction. RPA is always chromatin bound and present an speckled pattern regardless of the presence of DNA

damage, more especially in replicating cells. Usually, a preextraction step is required to visualize DNA-damage induced foci, but this is not compatible with Cyclin A staining, as the protein will diffuse away. I miss pictures of the cells untreated with CPT in order to be sure that the authors are really looking at damage-induced foci. Alternative resection can be visualized with a native BrdU immunofluorescence after incubating the cells for 24h with this analogue or alternative ways to follow cell cycle progression can be used (CENPF for example).

If the authors can clarify those points, I will be happy to support the publication of the manuscript in Nat Comm.

Reviewer #3 (Remarks to the Author):

General comments

The authors identify here a cycle of MRE11 sumoylation (SUMO2) and desumoylation catalized mainly by PIAS1 and SENP3, respectively, which operates in response to DSB-inducing agents and is critical for efficient DNA end resection and DSB repair. They made the initial interesting finding that suppression of SUMO2/3-ylation is detrimental for DNA end resection upon inhibition of topoisomerase I. Sumoylation occurs primarily on four major specific lysine residues on chromatin-bound MRE11, some of which have been linked to clinical variants leading to cancer, and its main role is to protect MRE11 from premature ubiquitin-related degradation on chromatin, barely affecting its interaction with NBS1 and RAD50 or its catalytic activity. The finding that sumoylation enhances nuclease activity and regulates end resection by antagonizing ubiquitin driven degradation of MRE11 is novel in the field and deserves appreciation.

This study also conveys further questions that warrant future research. One open question is the mechanism regulating the coordination between the desumoylating enzyme (SENP3) and a sensor of the extension of ssDNA, which would turn off the nuclease. A second issue is that the role of NBS1 sumoylation which, contrary to Mre11, decreases after CPT and other DSB-inducing agents. Although an interesting event by itself in terms of MRN complex regulation, this finding was not pursued in this study.

Although this referee is inclined to publish this study, a number of important issues in the conclusions should be resolved beforehand.

Major points

1. Figure 2 identifies PIAS1 as the main E3-ligase constitutively increasing sumoylation level of MRE11. However, overexpression of wt PIAS1 seems unable to further enhance MRE11 sumoylation in response to CPT (Figures 2b, 2c and 3d), therefore it could be argued that it is not specific to DSB. While PIAS1 knockdown resulted in a decreased CPT-induced MRE11-sumoylation, however the overall MRE11-SUMOylation was already diminished without CPT treatment. Likewise, the in vitro sumoylation assays with or without overhanging DNA and the interaction assays (Figure 2e-j) clearly show that MRE11 is indeed a substrate for PIAS1, but this interaction/ sumoylation is not shown to be responsive to CPT treatment by these experiments. Contrary to PIAS1, PIAS4 does appear to respond specifically to CPT treatment (Figure 2b vs Figure 2a) suggesting that it may have more specific roles or redundancy with PIAS1, but for some reason PIAS4 has been neglected. The reference cited (21) highlights a role for both PIAS1 and PIAS4 in DSB. The authors should clarify whether PIAS4 has (or not) more specific activity towards MRE11 than PIAS1. In addition, the authors should show whether the MRE11 sumoylation by PIAS1 (and PIAS4) occurs exclusively in the chromatin fraction to be able

to claim "MRE11 SUMOylation may be enhanced after its recruitment to chromatin" (page 4, lane 1).

2. The four potentially sumoylated lysines in MRE11 promote high molecular weight species and longer half-life in untreated conditions, the latter effect being exacerbated by CPT. In a similar scenario (Figure 4a), denatured purification of MRE11 followed by detection of ubiquitin detects a higher degree of ubiquitination in the MRE11-4KR variant (Figure 4b). As this result is key to understand the interplay between sumoylation/ ubiquitination in MRE11, could the authors explain why the Flag (=MRE11) signal from unmodified protein (≈ 100 kDa) is so different in input and in IP samples of the 4KR variant, even in untreated condition?
3. Cell viability assays in cells with endogenous MRE11 knocked down showed the effect of 0.5 or 1 microM CPT can be rescued by ectopically expressing MRE11 wt but not the MRE11-4KR variant (Figure 5d). However, CPT-concentration/viability does not match with those in figures 5e-f, where much lower concentrations appear to even intensify the viability loss.
4. Surprisingly, addition of a 3xpolySUMO2 chain fused at MRE11 is able to restore DNA resection (read as pRPA2), lower ubiquitination levels and cell viability to CPT and cisplatin to a MRE11-4KR variant. The explanation given is "the 3xSUMO2 chain may cover a wider surface to prevent ubiquitination conjugation to MRE11" (pg 7, lanes 20-21) is far from convincing. Could the authors speculate or, even better, provide alternative data that may help to explain this phenomenon? Eg, but not necessarily, by recruitment of additional proteins that would block ubiquitin E3-ligases, or intramolecular interaction with SIM motifs. This could be combined with a discussion as to how the four sumoylation sites on MRE11, which are fairly distant in the primary structure, have all similar and additive roles.
5. Data suggest that SENP3 may not be the only SENP acting on MRE11 nor specific to DSB, as siSENP3 has similar effect on MRE11 in +/- CPT conditions (Figure 7a and supplementary Figure 6a). Furthermore, CPT does not increase SENP3-MRE11 interaction (supplementary Figure 6c). This suggests that SENP3 may be a general recycling mechanism rather than a specific one. Based on the results presented, we still don't know how MRE11 is released into nucleoplasm for SENP3 to act on.
6. Based on the results of figures 7f-g, the authors concluded that "SENP3 desumoylates MRE11 primarily in the nucleoplasm instead of on chromatin" (pg 8, lane 9-10). However, this inference is in conflict with results in figure 7h, showing that SENP3 knockdown actually prevents MRE11 foci resolution, and with the conclusion of "[deSUMOylation by SENP3]...prevents its [MRE11] excessive accumulation on chromatin" (page 8, lanes 18-20).
7. In addition, the suggested model where SENP3 desumoylates MRE11 and consequently allows MRE11 ubiquitination and degradation, predicts that siSENP3 would produce excessive ssDNA and phosphor-RPA. In addition, MRE11 retention on damaged DNA, the authors should also show ensuing excessive ssDNA. Moreover, excess RPA on ssDNA is included in the model in figure 9, but actually not demonstrated.

Minor points

1. Figure 1A: Is the type of extract blotted WCE or Chromatin? What the effect in Sumo1?
2. Figure 1B: Time of treatment with 1 uM CPT? 45 min as in c?
3. Figure 1I shows that endogenous Mre11 purified with anti-Mre11 antibody is covalently attached to Flag-SUMO2. While this approach is convincing, a reciprocal IP with anti-Flag and detection of Mre11 is also desirable.
4. The nature of plasmids HA-K48 and HA-K63 used in Supplementary Figure 4a is not clear but the conclusions only make sense if they are ubiquitin K48-or K63-only lysine variants, respectively. This should be clearer in the manuscript.
5. The statement in Page 5, lines 17-19 does not match with the results in Figure 3d, which does show a small increment in sumoylation in the 4KR mutant after DNA damage and PIAS1 overexpression. Page 5, line 18. Change "...in 4KR, which" to "...in 4KR, it".
6. Page 5, line 33-4, remove 'with' before over-expressing and add 'the' before 4KR. Remove 'But', as the next sentence is not opposing the previous
7. Page 6, line 7: remove 'was'
8. Page 6, line 12: add 'was' at beginning on line.
9. 'S' and 'E' labels in figure 5g should be described in Figure legend.

10. Page 8, line 21: "nonspecific nucleolytic attacks". Could the authors explain better?
11. Figure 8a: showing the frequency of the clinical variants they identified at the sumo sites would allow inferring the relevance of these.
12. Page 10, line 4-6. How SENP3 may "play a role in maintaining MRE11 SUMOylation level on chromatin" should be explained better.

Our point-by-point response (using highlighting in blue):

Reviewer #1 (Remarks to the Author):

The manuscript from Zhang and colleagues characterized a novel post-translational regulatory mechanism of Mre11 upon DNA damage. They showed that Mre11 is SUMOylated upon DSB induction and is mediated by the SUMO E3 ligase PIAS1. Moreover, they found that SUMOylation of Mre11 is able to prevent it from ubiquitin-mediated degradation during DSB resection. In cells, expression of SUMOylation defective Mre11 led to defective-end resection and compromised homologous recombination. Overall, the data are convincing and are largely consistent with the authors' model. It should be well accepted by the genome research field. Below are a few additional comments that can help the authors to clarify and strength this already nicely performed work.

We are grateful to the reviewer for making helpful points to improve our manuscript and for his or her supportive comments.

1. Virtually all evidence for Mre11 SUMO modification was gathered while overexpressing His-tagged or Flag-tagged SUMO. Do the authors have evidence of DSB-induced modification of endogenous Mre11 by endogenous SUMO?

Thank you for this suggestion. The new data have been provided as **Supplementary Fig. 1g**. The endogenous MRE11 can be modified by endogenous SUMO2, and this SUMOylation is enhanced by CPT.

2. Can the interaction between Mre11 and PIAS1/SEN3 be regulated by DNA damage-inducing agents such as CPT and etoposide?

We thank the reviewer for this insightful suggestion. The new data have been provided in **Fig. 2f** and **Supplementary Fig. 6d**. These co-IP results showed that the PIAS1-MRE11 or SEN3-MRE11 interaction was not affected by DNA damage-inducing agent. Theoretically, MRE11 and PIAS1 are recruited to the damaged chromatin after DNA damage, thus, MRE11 should interact with more PIAS1 than that under undamaged conditions. However, to fully extract chromatin proteins, we sonicated the cell samples before incubation with the indicated beads. The PIAS1-DNA-MRE11 complex could be disrupted by sonication and recombine during incubation. Thus, co-IP may only reflect MRE11 and PIAS1/SEN3 interactions in cell lysate but not those specifically at DSB sites. In order to avoid misreading the result of experiment, we did not show **Fig. 2f** in the former manuscript.

3. The authors suggest that SUMOylation of Mre11 prevents it from ubiquitin-mediated degradation when resecting damaged chromatin. The half-life of wild-type Mre11 and its 4KR mutant should be determined in the presence/absence of PIAS1/SEN3.

We thank the reviewer for this important suggestion. The new data are provided in

Supplementary Fig. 4b and Supplementary Fig. 6c. The half-life of MRE11 protein is highly connected with PIAS1/SEN3-mediated SUMOylation/deSUMOylation.

4. The authors should use the well-established HR reporter assay (measured by the DR-GFP) to determine the impact of Mre11 SUMOylation on homologous recombination repair.

We thank the reviewer for this helpful suggestion. We performed the DR-GFP reporter assay and the result is now provided in **Fig. 5d**. The HR efficiency of the 4KR group was significantly lower than that of the WT group.

Reviewer #2 (Remarks to the Author):

In this manuscript, the authors describe a dynamic post-translational modification of the nuclease MRE11 in response to DNA damage. Not only they are able to describe this phenomenon, but also its relevance for DNA resection and survival to genotoxic agents and the SUMOylated residues of the protein. Furthermore, they are able to describe both the SUMO E3 ligase (PIAS1) and the SUMO deconjugated enzyme (SEN3) involved in the process. Finally, they can propose a mechanism in which transient sumoylation of MRE11 prevents its ubiquitylation, protecting the protein from degradation during a temporal window to allow resection. Later, SEN3-mediated SUMO deconjugation facilitates MRE11 removal in order to avoid hyper-resection. The relevance of this process is shown as mutations that impair this SUMOylation are found in cancer samples. Overall, the authors present a compelling story that is of interest of the field. In general, the data strongly support the conclusions of the authors, results are well presented and the manuscript is a solid candidate for publication in Nat Comm. It is true that opens interesting questions, such as how the temporal dynamic SUMOylation/deSUMOylation upon DNA damage is achieved as it looks ATM/ATR independent. Although it looks as it relies on a change on the nuclear distribution of SEN3, how this is achieved is not clear. In any case, in my opinion the advance this manuscript presents is enough at this stage to grant publication, leaving those open questions for further follow up analysis.

We sincerely appreciate the reviewer's positive comments and support. We admit this study's limitations such as how SEN3 senses DNA damage and precisely regulates MRE11 deSUMOylation remain unclear. We speculate that some signaling molecules, such as ssDNA fragments from DNA end resection or unknown proteins, may enter nucleolus and send signals to SEN3 after DNA damage. These questions are still unknown by now and worth further investigation.

Having said that, I have a grave concern about the in vivo analysis of resection that has to be clarified in order to get my full support for its publication. RPA foci is analyzed in S/G2 cells marked as Cyclin A positive upon CPT treatment. I have two important issues to mention:

1. Most Cyclin A positive cells will be in G2, especially those that show a strong staining. But CPT will only create breaks in S phase cells. Thus, most cyclin-positive cells should not show RPA foci. From the data presented it does not look like that. Either the text is misleading or

there is something strange on the experiment and G2 cells are indeed suffering chromosome breaks.

We thank the reviewer for making these comments. CPT-induced DNA breaks, as the reviewer mentioned, indeed only occur in S phase cells. The explanation for DNA breaks also seen in G2 phase cells could be that the S phase DNA breaks carry over into G2 phase. As for the S/G2 phase marker, Cyclin A starts to be expressed at the end of the G1 phase and persistently rises during the S phase, and is most active in the G2 phase, finally degrading in the early stage of the M phase (Ding et al, 2020, *Int J Mol Sci*; Mateo et al, 2010, *Biochem Soc Trans*; Yam et al, 2002, *Cell Mol Life Sci*). Analyzing pRPA2 foci in Cyclin A-positive cells is a method used in the field to assess DNA end resection (Bai et al, 2019, *Mol Cell*; Xu et al, 2020, *Oncogene*). We think Cyclin A is a relatively suitable S/G2 phase marker compared with Cyclin D, Cyclin E, and Cyclin B, although Cyclin A is not a precise marker for the S/G2 phase. We will try more accurate S/G2 phase markers in future work. And we also performed an experiment to confirm that CPT-induced pRPA2 foci were only seen in Cyclin A-positive cells (see below, **Fig. R1**).

2. Technically, I do not understand how the authors can visualize RPA foci without pre-extraction. RPA is always chromatin bound and present a speckled pattern regardless of the presence of DNA damage, more especially in replicating cells. Usually, a preextraction step is required to visualize DNA-damage induced foci, but this is not compatible with Cyclin A staining, as the protein will diffuse away. I miss pictures of the cells untreated with CPT in order to be sure that the authors are really looking at damage-induced foci. Alternative resection can be visualized with a native BrdU immunofluorescence after incubating the cells for 24h with this analogue or alternative ways to follow cell cycle progression can be used (CENPF for example).

If the authors can clarify those points, I will be happy to support the publication of the manuscript in Nat Comm.

Thank the reviewer for these insightful suggestions and questions. We performed the pRPA2 foci staining experiment with/without pre-extraction combined with DMSO/CPT treatment (see below, **Fig. R1**) and indeed observed the pRPA2 foci without pre-extraction (at least for pRPA2 antibody, Abcam, 87277). pRPA2 foci were totally CPT-induced with pre-extraction and without pre-extraction conditions, only that immunofluorescence pictures with pre-extraction had a clearer staining background. All pRPA2 foci were in Cyclin A-positive cells. However, as the reviewer pointed out, Cyclin A staining was not resistant to pre-extraction. Therefore, we performed these experiments without pre-extraction to observe pRPA2 and Cyclin A at the same time. Native BrdU immunofluorescence, another common method for assessing DNA end resection, and other alternative ways are all helpful suggestions and worth trying in our future studies.

Fig. R1 CPT-induced pRPA2 foci formation in Cyclin A-positive cells. HeLa cells were treated with 1 μ M CPT for 1 h, followed by immunostaining with the indicated antibodies. Scale bar, 50 μ m.

References

- Ding L, Cao J, Lin W, Chen H, Xiong X, Ao H, Yu M, Lin J, Cui Q. (2020). The Roles of Cyclin-Dependent Kinases in Cell-Cycle Progression and Therapeutic Strategies in Human Breast Cancer. *Int J Mol Sci*, 21, 1960.
- Mateo F, Vidal-Laliena M, Pujol MJ, Bachs O. (2010). Acetylation of cyclin A: a new cell cycle regulatory mechanism. *Biochem Soc Trans*, 38, 83-86.
- Yam CH, Fung TK., Poon RY. (2002). Cyclin A in cell cycle control and cancer. *Cell Mol Life Sci*, 59, 1317-1326.
- Bai Y, Wang W, Li S, Zhan J, Li H, Zhao M, Zhou XA, Li S, Li X, Huo Y, Shen Q, Zhou M, Zhang H, Luo J, Sung P, Zhu WG, Xu X, Wang J. (2019). C1QBP promotes homologous recombination by stabilizing MRE11 and controlling the assembly and activation of MRE11/RAD50/NBS1 complex. *Mol Cell*, 75, 1299-1314.
- Xu Z., Li X, Li H, Nie C, Liu W, Li S, Liu Z, Wang W, Wang J. (2020). Suppression of DDX39B sensitizes ovarian cancer cells to DNA-damaging chemotherapeutic agents via destabilizing BRCA1 mRNA. *Oncogene*. 39, 7051-7062.

Reviewer #3 (Remarks to the Author):

General comments

The authors identify here a cycle of MRE11 sumoylation (SUMO2) and desumoylation catalyzed mainly by PIAS1 and SENP3, respectively, which operates in response to DSB-inducing agents and is critical for efficient DNA end resection and DSB repair. They made the initial interesting finding that suppression of SUMO2/3-ylation is detrimental for DNA end resection upon inhibition of topoisomerase I. Sumoylation occurs primarily on four major specific lysine residues on chromatin-bound MRE11, some of which have been linked to clinical variants leading to cancer, and its main role is to protect MRE11 from premature ubiquitin-related degradation on chromatin, barely affecting its interaction with NBS1 and RAD50 or its catalytic activity. The finding that sumoylation enhances nuclease activity and regulates end resection by antagonizing ubiquitin driven degradation of MRE11 is novel in the field and deserves appreciation.

This study also conveys further questions that warrant future research. One open question is the mechanism regulating the coordination between the desumoylating enzyme (SENP3) and a sensor of the extension of ssDNA, which would turn off the nuclease. A second issue is that the role of NBS1 sumoylation which, contrary to Mre11, decreases after CPT and other DSB-inducing agents. Although an interesting event by itself in terms of MRN complex regulation, this finding was not pursued in this study.

Although this referee is inclined to publish this study, a number of important issues in the conclusions should be resolved beforehand.

We thank the reviewer for the valuable and positive comments on our manuscript. We are grateful to the reviewer for pointing out two interesting questions worth future research. One is how SENP3 and a sensor of ssDNA coordinate. It may be linked to that some signaling molecules, such as ssDNA fragments from end resection or unknown proteins sensing ssDNA extension, may enter nucleolus and send signals to SENP3. But the specific mechanism is unknown. Another is that NBS1 SUMOylation trends are contrary to MRE11. At the beginning of this study, we first explored the SUMOylation functions in the initiation of end resection. Because NBS1 SUMOylation decreasing after DNA damage is difficult to explain the observation that DNA end resection is inhibited by ML792, we did not follow this phenomenon in this study. However, as an interesting question, how NBS1 SUMOylation regulates resection is worth further investigation.

Major points

1. Figure 2 identifies PIAS1 as the main E3-ligase constitutively increasing sumoylation level of MRE11. However, overexpression of wt PIAS1 seems unable to further enhance MRE11 sumoylation in response to CPT (Figures 2b, 2c and 3d), therefore it could be argued that it is not specific to DSB. While PIAS1 knockdown resulted in a decreased CPT-induced MRE11-sumoylation, however the overall MRE11-SUMOylation was already diminished without CPT treatment. Likewise, the in vitro sumoylation assays with or without overhanging DNA and the interaction assays (Figure 2e-j) clearly show that MRE11 is indeed a substrate for PIAS1, but this interaction/ sumoylation is not shown to be responsive to CPT treatment by these experiments. Contrary to PIAS1, PIAS4 does appear to respond specifically to CPT treatment

(Figure 2b vs Figure 2a) suggesting that it may have more specific roles or redundancy with PIAS1, but for some reason PIAS4 has been neglected. The reference cited (21) highlights a role for both PIAS1 and PIAS4 in DSB. The authors should clarify whether PIAS4 has (or not) more specific activity towards MRE11 than PIAS1. In addition, the authors should show whether the MRE11 sumoylation by PIAS1 (and PIAS4) occurs exclusively in the chromatin fraction to be able to claim “MRE11 SUMOylation may be enhanced after its recruitment to chromatin” (page 4, lane 1).

We thank the reviewer for these important comments and suggestions.

First, PIAS1 was able to further elevate CPT-induced MRE11 SUMOylation (**Fig. 2b**, lane 2 vs. lane 1; **Fig. 2c**, lane 5 vs. lane 2; **Supplementary Fig. 2a**, lane 5 vs. lane 2, lane 6 vs. lane 3; new **Fig. 3d**, lane 7 vs. lane 3). For the former **Fig. 3d** (see below, lane 7 vs. lane 3), as the reviewer mentioned, PIAS1 overexpression did not obviously further enhance MRE11 SUMOylation in response to CPT, which may be caused by relatively low PIAS1 transfection levels. Therefore, we repeated this experiment with higher PIAS1 overexpression levels, and the result is now shown in new **Fig. 3d**. Besides, we agree with the reviewer that PIAS1 is able to SUMOylate MRE11 in the absence of CPT, as revealed by the data in **Fig. 2**. But, in the presence of CPT, PIAS1 recruited onto damaged chromatin induces much higher levels of MRE11 SUMOylation.

Second, PIAS4 indeed has a minor effect (**Fig. 2b**, lane 6 vs. lane 1) on MRE11 SUMOylation compared with PIAS1. Meanwhile, to further confirm the effect of PIAS1 and PIAS4 on MRE11 SUMOylation, we also performed denaturing pull-down assay to compare SUMOylation of MRE11 in PIAS1 and PIAS4 overexpression cells side by side, and PIAS4-induced MRE11 SUMOylation was far below the PIAS1-mediated increase (new **Supplementary Fig. 2b**). Thus, we mainly focus on PIAS1 in this study.

Third, according to the reviewer’s suggestion, we have added an experiment to detect PIAS1-mediated MRE11 SUMOylation after cell fraction extraction, and found that the majority of the PIAS1-induced SUMOylation was detected in chromatin fractions (new **Supplementary Fig. 2c**).

Former **Fig. 3d** SUMOylation of WT and 4KR under CPT treatment or PIAS1 overexpression was analyzed by IP with Ni-NTA beads and immunoblotting.

2. The four potentially sumoylated lysines in MRE11 promote high molecular weight species and longer half-life in untreated conditions, the latter effect being exacerbated by CPT. In a similar scenario (Figure 4a), denatured purification of MRE11 followed by detection of ubiquitin detects a higher degree of ubiquitination in the MRE11-4KR variant (Figure 4b). As this result is key to understand the interplay between sumoylation/ ubiquitination in MRE11, could the authors explain why the Flag (=MRE11) signal from unmodified protein (≈ 100 kDa) is so different in input and in IP samples of the 4KR variant, even in untreated condition?

Thank the reviewer for pointing out this question. In former and new **Fig. 4b**, owing to the overexpression of His-HA-Ub, both WT and 4KR were prone to be modified by Ub and degraded, even in CPT-untreated conditions (former and new **Fig 4b**, lanes 3&4 vs. lanes 1&2). After DNA damage, WT became more stable than that in CPT-untreated cells, but 4KR did not because of its SUMOylation defect (former and new **Fig 4b**, lane 5 vs. lane 3, lane 6 vs. lane 4). Compared to WT, unmodified 4KR protein levels were much lower in both treated and untreated conditions (former and new **Fig 4b**, lane 4 vs. lane 3, lane 6 vs. lane 5), which was due to that 4KR had much higher levels of ubiquitination. The degradation trends of WT/4KR in immunoblotting were affected by the transfection ratio of MRE11:ubiquitin, sample loading quantity, exposure time, degradation during IP incubation with beads, etc. We were also aware that the downward trend of the Flag signal in IP samples was larger than that in Input samples (former and new **Fig. 4b**). We conjectured that this experiment may be caused by either the further degradation during IP incubation with beads, or long exposure of the Flag panel in Input (former **Fig. 4b**). To further confirm this, we repeated this experiment and got a similar result

(new Fig. 4b).

Former Fig. 4b HEK293T cells expressing His-HA-Ub, SFB-WT, or SFB-4KR were treated with CPT or not. WT is more stable than 4KR.

3. Cell viability assays in cells with endogenous MRE11 knocked down showed the effect of 0.5 or 1 microM CPT can be rescued by ectopically expressing MRE11 wt but not the MRE11-4KR variant (Figure 5d). However, CPT-concentration/viability does not match with those in figures 5e-f, where much lower concentrations appear to even intensify the viability loss.

We are sorry for not making this clear in the Figure legend. For CCK-8 assay (former Fig. 5d, now Fig. 5e), cells were treated with 0.5 μ M or 1 μ M CPT for 36 h. For the colony formation assay (former Fig. 5e-f, now Fig. 5f-g), cells were treated with 5 nM or 10 nM CPT for 24 h, and after 14 days, cell survival was detected. In our preliminary experiments, treatment with 0.5 μ M or 1 μ M CPT for 24 h killed all cells after 14 days. Therefore, we used much lower concentrations of CPT for the colony formation assay. We now have revised the Fig. 5 legend.

4. Surprisingly, addition of a 3xpolySUMO2 chain fused at MRE11 is able to restore DNA resection (read as pRPA2), lower ubiquitination levels and cell viability to CPT and cisplatin to a MRE11-4KR variant. The explanation given is “the 3xSUMO2 chain may cover a wider surface to prevent ubiquitination conjugation to MRE11” (pg 7, lanes 20-21) is far from convincing. Could the authors speculate or, even better, provide alternative data that may help to explain this phenomenon? Eg, but not necessarily, by recruitment of additional proteins that would block ubiquitin E3-ligases, or intramolecular interaction with SIM motifs. This could be combined with a discussion as to how the four sumoylation sites on MRE11, which are fairly distant in the primary structure, have all similar and additive roles.

We thank the reviewer for these thoughtful suggestions and we totally agree with the reviewer. The fusion with a 3xSUMO2 chain rescues the phenotype of 4KR. Besides that 3xSUMO2 chain could block ubiquitin conjugation to MRE11, as the reviewer speculated, other explanations could be 3xSUMO2 chain blocks MRE11 interaction with certain E3 ubiquitin ligases (the main E3 ubiquitin ligases of MRE11 are still unknown so far), or 3xSUMO2 chain recruits certain SIM-containing proteins to prevent 4KR ubiquitination. We now have all these possible explanations discussed in the text (Page 7, lanes 28-31). In addition, although the four SUMOylation sites on MRE11 are distant in the primary structure, they may be close in the 3D structure and function together. However, as the full-length human MRE11 structure remains unsolved, this speculation still needs further to be confirmed.

5. Data suggest that SENP3 may not be the only SENP acting on MRE11 nor specific to DSB, as siSENP3 has similar effect on MRE11 in +/- CPT conditions (Figure 7a and supplementary Figure 6a). Furthermore, CPT does not increase SENP3-MRE11 interaction (supplementary Figure 6c). This suggests that SENP3 may be a general recycling mechanism rather than a specific one. Based on the results presented, we still don't know how MRE11 is released into nucleoplasm for SENP3 to act on.

We thank the reviewer for these comments.

First, among SENP1/2/3/5/6/7, we think SENP3 is the only SENP for the deSUMOylation of MRE11 at endogenous levels, as other SENPs' knockdown did not show an obvious effect (**Fig. 7a, Supplementary Fig. 6a**). And we agree with the reviewer that SENP3-mediated MRE11 deSUMOylation is not specific to DSB. Likewise, SENP3-MRE11 interaction was not affected by CPT treatment, as suggested by former **Supplementary Fig. 6c** (now **Supplementary Fig. 6d**). These data are in accord with our opinion that SENP3 plays a constitutive function in the deSUMOylation of MRE11, and SENP3 mainly deSUMOylates MRE11 in the nucleoplasm, and after DNA damage, more SENP3 is released into the nucleoplasm from the nucleolus. If excessive SUMO-MRE11 accumulates in the nucleoplasm, it may be re-recruited on the damaged chromatin and resect genome DNA. Thus, the main function of SENP3 is to prevent excessive SUMO-MRE11 accumulation in the nucleoplasm.

Second, for the question of "how MRE11 is released into nucleoplasm for SENP3 to act on", the explanation could be that MRE11 resects DNA from the endonucleolytic cleavage sites toward the DSB ends through its 3'-5' exonuclease activity and is finally released into the nucleoplasm from resected DSB ends by certain undefined mechanisms (Cejka and Symington, 2021, *Annu Rev Genet*; Reginato and Cejka, 2020, *DNA Repair (Amst)*). In the meantime, SENP3 is released into the nucleoplasm from the nucleolus (**Fig. 7f**) and acts on MRE11.

6. Based on the results of figures 7f-g, the authors concluded that "SENP3 desumoylates MRE11 primarily in the nucleoplasm instead of on chromatin" (pg 8, lane 9-10). However, this inference is in conflict with results in figure 7h, showing that SENP3 knockdown actually prevents MRE11 foci resolution, and with the conclusion of "[deSUMOylation by

SENP3]...prevents its [MRE11] excessive accumulation on chromatin” (page 8, lanes 18-20).

Thank you for making these comments. We think SENP3 deSUMOylates MRE11 primarily in the nucleoplasm. As for why SENP3 knockdown prevents the resolution of MRE11 foci on chromatin (**Fig. 7h**), our explanation is that MRE11 released into the nucleoplasm after end resection, if not deSUMOylated by SENP3 in time, will re-load onto damaged chromatin. Therefore, SENP3 can prevent MRE11 excessive accumulation on chromatin.

7. In addition, the suggested model where SENP3 desumoylates MRE11 and consequently allows MRE11 ubiquitination and degradation, predicts that siSENP3 would produce excessive ssDNA and phosphor-RPA. In addition, MRE11 retention on damaged DNA, the authors should also show ensuing excessive ssDNA. Moreover, excess RPA on ssDNA is included in the model in figure 9, but actually not demonstrated.

We thank the reviewer for this suggestion. pRPA2 level can represent ssDNA produced by DNA end resection. Thus, we tested pRPA2 foci in SENP3-knockdown cells and the new data are provided in **Supplementary Fig. 6f**. siSENP3 exhibited an over-resection phenotype after CPT treatment.

Minor points

1. Figure 1A: Is the type of extract blotted WCE or Chromatin? What the effect in Sumo1?

Thanks for these questions. The type of extract blotted in **Fig. 1a** is WCE, we now have added this information in the Figure legend. ML792 is a SUMO-activating E1 enzyme inhibitor. Therefore, all SUMO1/2/3-mediated modifications can be blocked. We also performed experiment to further confirm the ML792 effect on SUMO1 (**Supplementary Fig. 1b**).

2. Figure 1B: Time of treatment with 1 uM CPT? 45 min as in c?

In **Fig. 1b**, the time of CPT treatment is 1 h. We now have added more details to the Figure legend (Page 22, lanes 3-4).

3. Figure 1l shows that endogenous Mre11 purified with anti-Mre11 antibody is covalently attached to Flag-SUMO2. While this approach is convincing, a reciprocal IP with anti-Flag and detection of Mre11 is also desirable.

We thank the reviewer for this suggestion. For these experiments, we usually used His-SUMO2 instead of Flag-SUMO2 to pull down MRE11 under guanidine denaturing conditions, because guanidine buffer is better than SDS buffer to get rid of nonspecific modifications and Flag beads are not resistant to guanidine buffer. As shown in **Fig 1e, g-h**, we performed IP with His-SUMO2, and then detected SFB-MRE11. These data showed that His-SUMO2 can pull down MRE11.

4. The nature of plasmids HA-K48 and HA-K63 used in Supplementary Figure 4a is not clear

but the conclusions only make sense if they are ubiquitin K48-or K63-only lysine variants, respectively. This should be clearer in the manuscript.

We apologize for this confusing statement. Exactly as the reviewer mentioned, HA-K48 and HA-K63 are ubiquitin K48- or K63-only lysine variants (**Supplementary Fig. 4a**). We now use more clearer statements in the text (Page 6, lane 6), Figure and Figure legend (Page 25, lanes 34-35).

5. The statement in Page 5, lines 17-19 does not match with the results in Figure 3d, which does show a small increment in sumoylation in the 4KR mutant after DNA damage and PIAS1 overexpression. Page 5, lane 18. Change "...in 4KR, which" to "...in 4KR, it".

Thank the reviewer for pointing out these. According to the reviewer's suggestion, we now have corrected these in the text (Page 5, lines 23-24).

6. Page 5, line 33-4, remove 'with' before over-expressing and add 'the' before 4KR. Remove 'But', as the next sentence is not opposing the previous

Thank the reviewer for pointing out these. This sentence has been revised as suggested (Page 6, lines 2-4).

7. Page 6, line 7: remove 'was'

We have corrected this (Page 6, line 15).

8. Page 6, line 12: add 'was' at beginning on line.

Thank the reviewer for pointing out this, the sentence has been corrected (Page 6, line 20).

9. 'S' and 'E' labels in figure 5g should be described in Figure legend.

We thank the reviewer for pointing out this. 'S' and 'E' labels have been described in **Fig. 5g** legend (Page 23, lanes 37-38).

10. Page 8, line 21: "nonspecific nucleolytic attacks". Could the authors explain better?

We thank the reviewer for this helpful suggestion. This statement has been modified (Page 8, lines 30-31). Failing to deSUMOylate MRE11 by SENP3 results in genome instability due to excessive DNA end resection by redundant MRE11. It has been reported that inhibition of MRE11 degradation will cause excessive DNA end resection by MRE11 nuclease activity (Shenoy et al, 2019, *Cell*; Kilgas et al, 2021, *Cell Rep*).

11. Figure 8a: showing the frequency of the clinical variants they identified at the sumo sites would allow inferring the relevance of these.

Thank you for this suggestion. MRE11 K255E mutation has been found in uterine endometrioid carcinoma, ataxia-telangiectasia-like disorder and hereditary cancer-predisposing syndrome, and K384Q/I mutations have been found in hereditary cancer-predisposing syndrome. In the cBioPortal database, we found 1 of 399 patient samples of uterine endometrioid carcinoma with K255E mutation. For K384Q/I mutations, we did not find their patient mutation frequency in the database. Thus, we only show K255E patient mutation frequency in the text (Page 9, line 1).

12. Page 10, line 4-6. How SENP3 may “play a role in maintaining MRE11 SUMOylation level on chromatin” should be explained better.

Thank the reviewer for pointing out this. We realize several sentences here were inappropriate and confusing. We now have these modified in the text (Page 10, lines 15-17).

References

Cejka P, and Symington LS. (2021). DNA End Resection: Mechanism and Control. *Annu Rev Genet*, 55, 285-307.

Reginato G, and Cejka P. (2020). The MRE11 complex: A versatile toolkit for the repair of broken DNA. *DNA Repair (Amst)*, 91-92, 102869.

Jachimowicz RD, Beleggia F, Isensee J, Velpula BB, Goergens J, Bustos MA, Doll MA, Shenoy A, Checa-Rodriguez C, Wiederstein JL., Baranes-Bachar K, Bartenhagen C, Hertwig F, Teper N, Nishi T, Schmitt A, Distelmaier F, Ludecke HJ, Albrecht B, Kruger M, Schumacher B, Geiger T, Hoon DSB., Huertas P, Fischer M, Hucho T, Peifer M, Ziv Y, Reinhardt HC, Wiczorek D., Shiloh Y. (2019). UBQLN4 represses homologous recombination and is overexpressed in aggressive tumors. *Cell*. 176, 505-519.

Kilgas S. Singh AN, Paillas S, Then CK, Torrecilla I, Nicholson J, Browning L, Vendrell I, Konietzny R, Kessler BM, Kiltie AE, Ramadan K. (2021). p97/VCP inhibition causes excessive MRE11-dependent DNA end resection promoting cell killing after ionizing radiation. *Cell Rep*. 35, 109153.

REVIEWERS' COMMENTS

Reviewer #1 (Remarks to the Author):

The authors have addressed reviewers' comments by performing several experiments and the new data support authors' conclusions and strengthen the paper. I support the acceptance of the manuscript.

Reviewer #2 (Remarks to the Author):

The authors have clarified all my technical questions, thus I am happy to endorse the publication of the manuscript as it stands now.

Reviewer #3 (Remarks to the Author):

The authors have addressed the majority of my comments and I am happy with their revised version of the manuscript now.

Our point-by-point response (using highlighting in blue):

Reviewers' comments

Reviewer #1 (Remarks to the Author):

The authors have addressed reviewers' comments by performing several experiments and the new data support authors' conclusions and strengthen the paper. I support the acceptance of the manuscript.

Reviewer #2 (Remarks to the Author):

The authors have clarified all my technical questions, thus I am happy to endorse the publication of the manuscript as it stands now.

Reviewer #3 (Remarks to the Author):

The authors have addressed the majority of my comments and I am happy with their revised version of the manuscript now.

We appreciate the reviewers for their time to reevaluate our revised manuscript. We are very pleased that all the reviewers are satisfied with this revision and support the publication. We sincerely thank the reviewers again for making many good points to enhance the appeal of our manuscript further.